# Modeling vegetation and carbon dynamics of managed grasslands at the global scale with LPJmL 3.6

Susanne Rolinski[1], Christoph Müller[1], Jens Heinke[1], Isabelle Weindl[1,2,3], Anne Biewald[1], Benjamin Leon Bodirsky[1], Alberte Bondeau[4], Eltje R. Boons-Prins[5], Alexander F. Bouwman[6], Peter A. Leffelaar[5], Johnny A. te Roller[7], Sibyll Schaphoff[1], and Kirsten Thonicke[1]

[1]Potsdam Institute for Climate Impact Research (PIK), Member of the Leibniz Association, P.O. Box 60 12 03, D-14412 Potsdam, Germany
[2]Humboldt University of Berlin, Unter den Linden 6, 10099 Berlin, Germany
[3]Leibniz Institute for Agricultural Engineering and Bioeconomy, Max-Eyth-Allee 100, 14469 Potsdam, Germany
[4]Institut Méditerranéen de Biodiversité et d'Ecologie marine et continentale (IMBE), Aix Marseille Université, CNRS, IRD, Avignon Université, 13545 Aix-en-Provence cedex 04, France
[5]Wageningen University and Research, Plant Production Systems, Droevendaalsesteeg 1, 6708 PB Wageningen, Netherlands
[6]Department of Earth Sciences - Geochemistry, Faculty of Geosciences, Utrecht University, PO Box 80021, 3508 TA Utrecht, The Netherlands
[7]Alterra, Wageningen Environmental Research, PO Box 47, 6700 AA Wageningen, Netherlands

*Correspondence to:* Susanne Rolinski (susanne.rolinski@pik-potsdam.de)

**Abstract.** Grassland management affects the carbon fluxes of one third of the global land area and is thus an important factor for the global carbon budget. Nonetheless, this aspect has been largely neglected or underrepresented in global carbon cycle models. We investigate four harvesting schemes for the managed grassland implementation of the DGVM LPJmL that facilitate a better representation of actual management systems globally. We describe the model implementation and analyze simulation results with respect to harvest, net primary productivity and soil carbon content and by evaluating them against reported grass yields in Europe. We demonstrate the importance of accounting for differences in grassland management by assessing potential livestock grazing densities as well as the impacts of grazing, grazing intensities and mowing systems on soil carbon stocks. Grazing leads to soil carbon losses in polar or arid regions even at moderate livestock densities ($< 0.4$ LSU ha$^{-1}$) but not in temperate regions even at much higher densities (0.4 to 1.2 LSU ha$^{-1}$). Applying LPJmL with the new grassland management options enables assessments of the global grassland production and its impact on the terrestrial biogeochemical cycles, but requires a global data set on current grassland management.

# 1   Introduction

Managed grasslands and rangelands cover 25 % of the Earth's ice-free land surface (FAOSTAT, 2016) and 68 % of the agricultural area (Steinfeld et al., 2006). Their extent remains relatively stable over time with an increase from 3100 Mha in 1960 to 3400 Mha in 1995 and until now (FAOSTAT, 2016). The productivity of grasslands depends on climatic and soil conditions, as well as on the amount and frequency of biomass removal by mowing or herbivores (wild animals or livestock) (Pachzelt et al., 2013). There are high uncertainties with respect to the management of grasslands and the grazing intensity on pastures (Fetzel et al., 2017). Even in a recent study by Erb et al. (2016b), the uncertainty in the grazing area is given as ± 40 % in comparison to the average. Considering livestock density and fertilization as management intensity factors, less than 10 % of the grassland area estimated to be under *high*, about 65 % under *medium* and 25 % under *low* grazing intensity (Erb et al., 2016b). A better separation of grassland management into areas with grazing or mowing on the global scale has been identified as a major challenge for better assessments of land management (Erb et al., 2016b) partly because of conflicting definitions of landuse (Erb et al., 2007; Ramankutty et al., 2008).

## 1.1   Characteristics of managed grasslands

Agricultural area that is covered predominantly by grasses or other herbaceous forage plants for a duration of at least 5 years is classified as 'permanent pastures' or 'managed grassland' (Commission, 2004; Ramankutty et al., 2008). We use the term 'managed grassland' for these open landscape ecosystems with herbaceous vegetation which is mown or grazed by animals. We do not further distinguish (semi-)natural grasslands, as the distinction of the management intensity or the proportion between livestock and wild animals is mostly difficult (Ramankutty et al., 2008). Managed grassland differs from natural vegetation or cropland in the use of biomass to feed livestock mostly without additional irrigation or fertilization.

There are two mechanisms how management affects the vegetation composition of managed grasslands: 1. the establishment of plants that can be directly influenced by sowing of highly productive and nutritious grasses and 2. livestock grazing, as animals prefer some species which may then disappear under high grazing pressure (Brown and Stuth, 1993; Sharp et al., 2014). Both mechanisms are intentionally used in European livestock systems to maximize herbage digestibility by sowing suitable forage cultivars and frequent grazing (Soussana et al., 2004).

The management of grasslands plays an important role in global carbon and water cycles (Herrero et al., 2016). The frequency of biomass removal and its fate (complete removal as feed to other locations or partially remaining in the form of manure on the plot) have effects on the productivity of the grass itself and on the carbon and water budget of the grassland (Soussana et al., 2004; Herrero et al., 2016).

The form of harvesting and the amount of herbaceous biomass that is harvested are central elements of grassland management. For intensive grazing in Europe (about 2 LSU ha$^{-1}$), Soussana et al. (2004) report a harvest fraction of 60 % of the above-ground dry matter production and less for lower livestock densities. The digestible part of the ingested carbon (up to 75 %) is respired shortly after intake (Soussana et al., 2004). 25 - 40 % of the carbon intake is non-digestible and returned to the grassland in the form of excreta (Soussana et al., 2004). Thus, management includes also the possibility to apply manure

which is either directly dropped by livestock or spread by machinery. From the excreta that livestock directly release on grassland, globally only a small fraction is recovered for use outside the grassland ecosystem (Sheldrick et al., 2003) although local differences may be substantial. From the excreta produced in a stable, about 44 % (10 to 50 %) of the solid manure is allocated to pastures in developed countries whereas this portion is below 10 % in developing countries (Liu et al., 2010; Bouwman et al., 2002; Smil, 1999).

Mismanagement, on the other hand, can have deteriorating impacts. Overgrazing or trampling play a role especially in arid areas or under high livestock density (Dlamini et al., 2016). Mismanagement also plays a role in the increase in desertification and is the main reason of soil degradation of 15 % in the drylands in Sub-Saharan Africa (Kiage, 2013).

## 1.2 Representation of managed grasslands in DGVMs

Modeling grassland dynamics has a long tradition and a multitude of approaches (Chang et al., 2013, and references therein) were developed, but mostly applied at the plot scale. Dynamic Global Vegetation Models (DGVMs) provide a suitable modeling framework to also assess grassland dynamics, grassland productivity, and the impact on the biogeochemical cycles under different grassland management schemes at the global scale.

When the LPJ DGVM (Lund-Potsdam-Jena, Sitch et al., 2003) was expanded by agricultural activities, forming LPJmL (Lund-Potsdam-Jena managed Land, Bondeau et al., 2007), this extension included the integration of managed grassland. In that implementation, managed grassland was considered as grassland ecosystem with a harvesting rule. Grass plants on managed plots were simulated as natural grasses and competed for light and water. The frequency of harvest events was solely depending on grass productivity. When more than 100 gC m$^{-2}$ was assimilated since the last harvest event, half of the aboveground carbon was removed from the plot. Assimilated carbon was allocated to leaves and roots prior to harvest and at the end of the year following the allocation rules for natural grasses as described by Sitch et al. (2003).

An implementation of management techniques into a DGVM was presented for the Organizing Carbon and Hydrology in Dynamic Ecosystems model (ORCHIDEE, Chang et al., 2013) coupled with the plot-scale pasture model PaSim (Vuichard et al., 2007). While the implementation of grazing and mowing is demonstrated at the European scale, a recent application is combining satellite-derived productivity and model simulations at the global scale to derive historical changes in grassland management (Chang et al., 2016).

To the knowlegde of the authors, managed grasslands are not represented in further DGVMs. The Community Land Model (CLM) treats pasture in the version with a representation of agricultural activity (CLM-Crop, Drewniak et al., 2015) as natural grassland without harvest procedure. JSBACH (Reick et al., 2013) simulates fire disturbances on grassland but no other forms of biomass removal are taken into account.

## 1.3 Approach

We here extend the representation of managed grasslands in LPJmL by explicitly describing four different management options of herbaceous vegetation with the presence of livestock or the use of harvested grass as livestock feed. The annual sum of grass biomass which is removed from managed grasslands is referred to as harvest or yield. We define management options

by combining different biomass removal frequencies and amounts as well as the conversion of grass biomass to manure for grazing systems, resulting in the description of one default option and the implementation of three new management options. These new management options are designed to cover the range of different possible management schemes with respect to their characteristics in productivity, as well as in carbon and water dynamics. Besides a default management option $D$ similar to that in Bondeau et al. (2007), we add the following three grassland management options:

$M$: a regular mowing scheme as e.g. applied for the production of hay,

$G_D$: a continuous grazing system with flexible livestock densities, and

$G_R$: a rotation grazing system, in which ruminants of flexible densities are moved between individual paddocks in regular intervals.

Without being able to drive the model with data on actual grassland management patterns at this stage, we pursue the following objectives with this implementation:

- comprehensive representation of the diversity in grassland management and in related feedbacks between biomass removal and primary productivity,

- demonstration of the role of grassland management for biogeochemical simulations by analyzing the effects on grass yield, Net Primary Productivity (NPP) and soil carbon stocks,

- assessment of potentials of agricultural productivity by determining maximum harvest and the associated livestock densities with and without the condition of maintaining soil carbon stocks,

- evaluation of model performance by comparing simulated harvest with an European data set (Smit et al., 2008) and potential livestock densities with data from the Gridded Livestock of the World v2.0 (Robinson et al., 2014).

## 2 Methods

In this section we describe the data sources (2.1), an overview of the modeling concept (2.2), the representation of grass growth in LPJmL (2.3), the implementation of management options (2.4), the configuration of the model simulations (2.5) and the methodology of the analysis (2.6).

### 2.1 Data sources

Climate data for model simulations include monthly temperature and cloudiness from the Climate Research Unit's (CRU) time series (TS) 3.1 data (Mitchell and Jones, 2005) and monthly gridded precipitation from the Global Precipitation Climatology Centre's (GPCC, version 5) (Rudolf et al., 2010). Monthly climate data are interpolated by the model internally to daily values by linear interpolation for temperature and cloudiness and by an internal weather generator (Gerten et al., 2004) for precipitation

using the number of wet days as described by New et al. (2000). Global annual values for atmospheric $CO_2$ concentration are used from the Mauna Loa station (NOAA/ESRL, www.esrl.noaa.gov/gmd/ccgg/trends/). Thermal and hydraulic characteristics of the soils are derived from the Harmonized World Soil Database (version 1.2) (2012). These data were first aggregated to $0.5°$ resolution and classified according to the USDA soil texture classification (http://edis.ifas.ufl.edu/ss169) (Schaphoff et al., 2013).

For the evaluation of simulated yield potentials, average yield data for European subnational entities for the year 2000 were kindly provided by Smit et al. (2008). Simulation results were averaged over the corresponding geographical units and for the years 1995 to 2004 and aggregated to the subnational units. For the spatial aggregation, we computed area-weighted means per spatial unit, using the pasture area per grid cell as weights which was derived by Fader et al. (2010) by modifying the dataset of Portmann et al. (2010).

## 2.2 Overview of modeling concepts in LPJmL

LPJmL simulates carbon and water cycles as well as vegetation growth dynamics depending on daily weather conditions and soil texture. We refer to the current status of the model prior to the implementation of managed grasslands as LPJmL3.5. Simulations in this study are conducted on a regular grid at $0.5° \times 0.5°$, but as the model is essentially a point model, it can be run at any spatial resolution provided by the input data. The soil depth of 3 m is divided into 5 soil layers with thicknesses of 0.2, 0.3, 0.5, 1.0 and 1.0 m, respectively. The model calculates carbon fluxes (gross primary production, auto- and heterotrophic respiration) and the respective changes in carbon pools (leaves, sapwood, heartwood, roots, storage organs, litter and soil), as well as water fluxes (interception, percolation, evaporation, transpiration, snowmelt, runoff) (Gerten et al., 2004; Rost et al., 2008). Closed mass balances across all fluxes and pools are ensured for carbon and water while carbon and water pools adjust dynamically according to the in- and outgoing fluxes. Photosynthesis is simulated following a simplified Farquhar model approach for global simulations (Farquhar et al., 1980; Collatz et al., 1991, 1992; Haxeltine and Prentice, 1996a, b). After the implementation of agricultural land by Bondeau et al. (2007), grid cells are separated into different spatial units, called *stands*, with their specific carbon, water and energy budgets. Plant growth, vegetation dynamics and the associated water and carbon dynamics are simulated for representative average individuals of different plant functional types (PFTs). Natural PFTs grow on the same stand, competing for light and water (Sitch et al., 2003). Crops, on the other hand, are simulated on individual stands assuming mono-cultures. While managed grassland is simulated on separate stands as well, vegetation can still consist of up to two herbaceous PFTs (one $C_3$ and one $C_4$ herbaceous PFT) which compete for resources.

### 2.2.1 Natural vegetation

Natural vegetation is represented in LPJmL at the biome level by nine PFTs (Sitch et al., 2003). Processes of carbon assimilation and water consumption are parameterized on the leaf level and scaled to the simulation unit. Carbon assimilation by photosynthesis, water fluxes and plant and soil respiration are computed at daily time steps, whereas the allocation to the vegetation carbon pools is updated at annual time steps. Intra-annual dynamics of leaf area and, thus, light interception, are computed by scaling the leaf biomass with a phenology-dependent factor. Litterfall of leaves from vegetation upon mortality

or from tissue turnover accumulates in above- and below-ground litter pools. Decomposition from these litter pools feeds into soil carbon pools with fast (10-year turnover) and slow (100-year turnover) decomposition rates. Soil and litter decomposition is controlled by soil moisture and soil temperature. All soil processes are computed on a daily basis. For further details see Schaphoff et al. (2013). Carbon and water dynamics are linked so that the effects of changing temperatures, water availability and $CO_2$ concentrations are accounted for (Gerten et al., 2004, 2007). Physiological and structural plant traits of each PFT determine its water requirements and consumption.

Competition between PFTs due to differences in their performance under given climate conditions can lead to changes in vegetation composition. Changes in the PFT distribution in turn affect the productivity of individual PFTs in subsequent time steps, leading to changes in carbon and water fluxes. These fluxes are also impacted by the dynamics of the input data (weather data, soil), accounting for long-term climate trends, interannual climate variability and the impact of extreme events.

### 2.2.2 Agro-ecosystems

Plant growth and agricultural production on cropland is represented by 12 crop functional types (CFTs) as described in Bondeau et al. (2007) and Müller and Robertson (2014). Crops can be simulated as irrigated or fully rainfed production systems (Rost et al., 2008; Jägermeyr et al., 2015), each system dedicated to its own 'stand' (see 2.2 Modelling concepts) and with its own water budget so that irrigation water is not transferred to rainfed cropland. CFTs do not represent one specific cultivar, instead parameters that represent characteristics of a specific crop variety are internally selected depending on the local climate conditions for each CFT (Bondeau et al., 2007).

In contrast to PFTs in the natural vegetation, the allocation of assimilated carbon is simulated on a daily basis to better account for environmental impacts during different stages of crop growth and to better account for the actual growing period of annual crops. During fallow periods, crop stands are merged into a setaside stand, where soil properties (carbon, water) are mixed according to their spatial extent. Irrigated agricultural land is kept on a separate setaside stand to avoid irrigation water transfer to rainfed stands upon the next cultivation cycle. Newly sown crops are planted on stands that are initialized to the conditions of the setaside stand on that particular day.

## 2.3 Managed grassland

### 2.3.1 Overall setting

Managed grasslands are implemented as agricultural 'stands' (see 2.2 Modelling concepts). Establishment of herbaceous PFTs on managed grassland stands follows similar rules to those for natural vegetation, but woody PFTs (trees) are not allowed to establish. The herbaceous $C_3$ and $C_4$ PFTs can grow together and compete for light and water. Typically, just one PFT is present on managed grassland stands because the overlap of their bioclimatic limits is quite narrow. Parameter settings are chosen as in LPJmL3.5 unless denoted otherwise.

The main difference of managed grasslands to natural grasslands is the occurrence of harvest events, i.e. removal of leaves by mowing or grazing. We here describe the implementation of a daily allocation routine for the assimilated carbon in grass

plants which is a prerequisite for the three explicit management options as well as a default setting, which can be used in the absence of specific knowledge on management regimes or available input data sets.

### 2.3.2 Parametrization of daily allocation

Flexible harvest schemes on managed grasslands require that the allocation of assimilated carbon ($B_I$) occurs on a daily basis
as for crops (Section 2.2.2), rather than on an annual basis as in the implementation of natural vegetation (Section 2.2.1). Partitioning of assimilated carbon $B_I$ to leaves and roots is calculated such that a given leaf to root mass ratio is approximated. The PFT-specific parameter $lr_p$ is 0.75 for both grasses (Sitch et al., 2003), i.e. that leaf carbon equals 0.75 times root carbon under optimal conditions. $lr_p$ is scaled to the actual ratio $lr$ (Eq. 1) with a measure of water stress (actual ratio of plant water supply $W_{supply}$ to atmospheric water demand $W_{demand}$) (Eq. 2 in Sitch et al., 2003). Under dry conditions, this scaling results
in a lower $lr$ so that the allocation of assimilated carbon is shifted towards the roots.

$$lr = \max(0.25, lr_p \cdot \min(1, W_{supply}/W_{demand})) \tag{1}$$

On days with positive net primary productivity (NPP), i.e. when more carbon is assimilated than needed for maintenance respiration, the biomass increment $B_I$ is positive and allocated to the leaf carbon pool ($L$) and the root carbon pool ($R$) by calculating their increments ($L_I, R_I$) (Eqs. 2 and 3).

$$
\quad L_I = \min\left(B_I, \max\left(\frac{B_I + R - L/lr}{1 + 1/lr}, 0\right)\right), \tag{2}
$$
$$
R_I = B_I - L_I. \tag{3}
$$

When maintenance respiration outweighs carbon assimilation, NPP is negative. In this case total plant biomass is reduced. No reallocation from leaves to roots or vice versa is assumed, but the negative biomass increment $B_I$ is divided between leaves and roots proportionally to their biomass and both compartments are reduced by the increments ($L_I, R_I$).

After a harvest event, leaf carbon and thereby the actual leaf to root mass ratio is reduced. Carbon allocation in the following period will try to reestablish the actual leaf to root mass ratio $lr$. Depending on the water supply to demand ratio, the assimilated carbon is incorporated more to the leaves so that the actual water conditions and NPP determine the recovery time of the leaves. Without a feedback to primary productivity, a 10 % reduction of the water supply would result in a slower recovery time of several days (as more carbon would be allocated to the roots) and leaves would have less carbon (and thus less area
for intercepting light) when the new $lr$ is reached. Accounting for the reduced leaf area after harvest events, photosynthetic capacity and light interception are reduced as well so that harvest of leaf carbon induces a feedback on photosynthesis. Following the calculations and parameter settings as in Sitch et al. (2003), the actual leaf carbon $L$ and the specific leaf area (SLA in $m^2$ $gC^{-1}$, Eq. 4 with leaf longevity $\alpha_{leaf}$ in yr) are used to update leaf area index (LAI, Eq. 5) and foliage projected coverage (FPC in $gC^{-1}$, Eq. 6 with Lambert-Beer parameter $k_b$). With FPC, also the fraction of absorbed photosynthetic active radiation
(fAPAR, Eq. 7 with the number of plant individuals $N_{ind}$) is changed describing the part of radiation that is absorbed for photosynthesis. Some of the parameters have standard values such as $\alpha_{leaf} = 1$ for grasses, i.e. grass leaves are assumed to be photosynthetically active for 1 year (Eq. (6) in Sitch et al., 2003), $N_{ind} = 1$, i.e. that one average individual is considered (as

set in Sitch et al., 2003), and $k_b = 0.5$, as established in the literature and recently confirmed by Saitoh et al. (2012) (Eq. (7) in Sitch et al., 2003).

$$
\begin{aligned}
\text{SLA} &= 2 \cdot 10^{-4} \cdot \exp(6.15 - 0.46 \cdot \log(\alpha_{leaf} \cdot 12)) \ \text{ with } \ \alpha_{leaf} = 1 \\
&= 0.02988943 \tag{4}
\end{aligned}
$$

$$
\text{LAI} = L \cdot \text{SLA} \tag{5}
$$

$$
\begin{aligned}
\text{FPC} &= N_{ind} \cdot (1 - \exp(-k_b \cdot \text{LAI})) \ \text{ with } \ N_{ind} = 1 \ \text{ with } \ k_b = 0.5 \\
&= 1 - \exp(-0.5 \cdot \text{LAI}) \tag{6}
\end{aligned}
$$

$$
\text{fAPAR} = \text{FPC} \tag{7}
$$

This dependency of light absorption and photosynthesis on leaf carbon content leads to a negative feedback of harvest on absorbed radiation. When leaf carbon is reduced to 50 %, the reduction of fAPAR is about 30 % for $L = 100$ gC m$^{-2}$ and is diminished to 2 % for $L = 500$ gC m$^{-2}$.

## 2.4 Implementation of grassland management options

Pastures are managed through mowing, grazing or a combination of both, depending on the grassland productivity as well as on many other factors such as the availability of labor force. When mowing is not an option due to the steepness of the landscape, soil wetness or obstructing trees or boulders, grazing by small ruminants might still be possible but mowing and grazing by livestock are often used in combinations. Grazing with low densities of livestock over longer time periods or even the entire vegetation period is often combined with a few cuts and sometimes with more frequent mowing events. While most rangelands are not cut at all, grasslands can be grazed by high livestock densities for short time periods but rarely mowed. To avoid the implementation of a huge set of possible combinations of grazing with different livestock densities and mowing frequencies, we decided to choose four basic regimes. In absence of grazing animals, option $D$ (for default) represents frequent and option $M$ few mowing events. Without mowing, options $G_D$ and $G_R$ distinguish between permanent low density and rotational high density grazing. The following paragraphs provide detailed descriptions of the management options (Tab. 1) with respect to harvest frequency, livestock density, and parameters. For a discussion on parameter choices, we refer to section 4.2.

### 2.4.1 Frequent mowing without grazing – Default option $D$

For the default option $D$ no specific assumptions on grassland management are necessary. Its main purpose is to provide a generic accounting for biomass removal for livestock feed. Under the improved harvest scheme $D$, harvesting is possible at the end of each month ($H_{day}$), provided that the leaf biomass increment over all herbaceous PFTs since the last harvest event is positive.

Leaf biomass removal is based on harvest index $H_{frac}$ (-) (Fig. 1), which is calculated internally via a Michaelis-Menten function using the leaf carbon $L$ and half saturation constant $H_p = 1000$ (gC m$^{-2}$). In unproductive systems, only a small

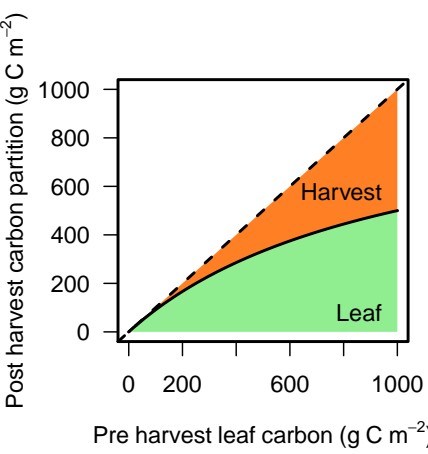

**Figure 1.** Partitioning of leaf carbon into harvested fraction (between solid and dashed lines) and remaining fraction (below solid line) depending on leaf carbon content before harvest (x-axis).

**Table 1.** Characteristics of grassland management options.

|  | $D$ | $M$ | $G_D$ | $G_R$ |
|---|---|---|---|---|
| Harvest |  |  |  |  |
| – frequency | biomass dependent | fixed | daily | daily |
| – period | last day of month | twice per year | during vegetation period | few days followed by recovery period |
| Soil feedback via |  |  |  |  |
| – roots & stubble | yes | yes | yes | yes |
| – manure | no | no | yes | yes |
| Livestock density | none | none | 0.5 LSU ha$^{-1}$ | 1.2 LSU ha$^{-1}$ |

fraction of total leaf biomass is removed, increasing with overall leaf biomass. At a leaf biomass of 1000 gC m$^{-2}$, half of the leaf carbon is harvested and removed entirely from the plot.

### 2.4.2 Few mowing events without grazing – Option $M$

The mowing option $M$ represents a regime with several mowing events per year. This option may be adjusted to local conditions by scheduling these events to certain dates or according to climatic conditions. For the global application, two harvest events per year are scheduled six months apart with one event on June 1$^{st}$ and another one on December 1$^{st}$. In so doing, the mowing option is identical on both hemispheres but can fail for events not within the growing period.

Each individual harvest event is only realized when the leaf carbon content exceeds the threshold amount of 25 gC m$^{-2}$. Leaf carbon above the threshold is harvested and removed entirely like for option $D$. After mowing, the remaining carbon content of the leaves is thus 25 gC m$^{-2}$ which does not necessarily correspond to a specific leaf height. LAI is computed from

the reduced leaf carbon according to Eq. 5. We acknowledge that there may be more mowing events than two in productive systems or with different timing, but assume no variation in mowing events for simplicity.

### 2.4.3 Daily grazing without mowing – Option $G_D$

When temperatures are sufficiently high to enable grass growth, i.e. above $5\,^\circ$C, a fixed portion of the leaf carbon $L$ is removed each day per grazing livestock unit (LSU) which is corresponding to a cow of 650 kg liveweight (Chesterton et al., 2006, based on EC definitions). The stocking density can be specified per grid cell or is set to a default value of $S_D = 0.5$ LSU ha$^{-1}$. To avoid permanent damage to the managed grassland, grazing is allowed only when a minimum threshold of 5 gC m$^{-2}$ of leaf carbon is present, assuming that the livestock is removed or fed externally during these low biomass periods.

Daily intake of carbon is varying between grazing animals and seasons and it was necessary to find a value that represents the demand for grass carbon for one livestock unit independent from the corresponding production system, i.e. independent from the amount of additional feed from other sources. Cordova et al. (1978) propose to estimate daily intake corresponding to the metabolic body weight (MBW = liveweight$^{0.75}$); in this case MBW $= 129 (= 650^{0.75})$ for the chosen LSU. The daily intake varies between 18 and 41 gC day$^{-1}$ MBW$^{-1}$ (40 to 90 gDM day$^{-1}$ MBW$^{-1}$) which gives a range of the daily intake as 2300 to 5200 gC day$^{-1}$ LSU$^{-1}$. For livestock in organic farming, Kristensen et al. (2011) give estimates for feed intake and the portion of pasture feed that result in 2100 gC day$^{-1}$ LSU$^{-1}$ on average (minimum 780 and maximum 3450 gC day$^{-1}$ LSU$^{-1}$). We assume the daily demand at 4000 gC LSU$^{-1}$ day$^{-1}$ (corresponding to 8.9 kg DM LSU$^{-1}$ day$^{-1}$) assuming that high productive livestock requires a certain portion of grass feed along with concentrates from other sources.

The carbon from grazed biomass is incorporated into the animals, transferred to the soil carbon pool as well as mineralized to $CO_2$. The portion of the grass intake that is remaining on the grassland as manure of 25 % (Soussana et al., 2014) is incorporated into the fast soil carbon pool. We do not distinguish the portions of the grazed carbon that is going to animals or to mineralization.

### 2.4.4 Rotational grazing without mowing – Option $G_R$

An alternative grazing management type represents a rotational system of grazing on several paddocks for a short duration and includes longer recovery periods after the grazing phase. For the implementation of this system in a grid-cell-based model, we define a number of rules on the subdivision of the grassland stand into a number of paddocks:

1. We simulate one of the paddocks and assume that the overall carbon and water budget of the other paddocks is similar, but with a temporal delay. The length of one rotation includes the grazing period at the beginning and the following recovery period.

2. The rotation period begins when the leaf carbon is above a threshold value of 40 gC m$^{-2}$ (based on recommendations e.g. in Williams and Hall, 1994; Undersander et al., 2002; Blanchet et al., 2003, see also section 4.2). The length of the rotation period $R_L$ in days is determined by the division of the current leaf carbon by the daily demand of the given livestock and restricting this number to a maximum value of 50 days (usually between 20 and 35 days). The

stocking density can be specified per grid cell or is set to a default value of $S_R = 1.2$ LSU ha$^{-1}$ and the daily demand of 4000 gC LSU$^{-1}$ day$^{-1}$ is the same as for option $G_D$.

3. For the determination of the number of paddocks, the rotation length $R_L$ is chosen between an upper limit of $uR_L = 50$ days and a lower limit of $lR_L = 1$ day. The maximum number of paddocks is $P_{max} = 16$.

4. The grazing period on one paddock ends when a minimum threshold of the remaining leaf carbon $L = 5$ gC m$^{-2}$ is reached which should represent a stubble height of 5 cm. Similar to option $G_D$, the removed carbon is divided into a harvest flux (incorporation into animal body and animal respiration) and a manure application to the soil pool which is subject to mineralization there. The portion of the grass intake that is remaining on the grassland as manure (which is incorporated into the fast soil carbon pool) of 25 % (Soussana et al., 2004).

**2.5  Set up of model runs**

The model runs on $0.5° \times 0.5°$ spatial resolution and with a daily time step. Spinup and transient runs are conducted using interpolated monthly climate data (section 2.1) from 1901 until 2009. Model simulations are based on a spin-up run of 5000 years with natural vegetation using monthly input (section 2.1) from 1901 until 1930 in repetitive loops. This spinup simulation is needed to bring potential natural vegetation composition into a dynamic equilibrium and then the corresponding soil carbon

pools. Long simulation cycles are necessary, especially because of the simulated permafrost dynamics where processes are slow and soil carbon stocks are large (Schaphoff et al., 2013). After the 5000-year spinup, a 390-year spinup is conducted to account for land-use change since 1700. Nearly twice the length of the landuse history of 200 years from 1700 to 1900 is needed to achieve consistent starting conditions for the transient simulations after the dynamic soil carbon equilibrium under potential natural vegetation has been disturbed.

As information on the global distribution of grassland management activities is lacking (Erb et al., 2016b), our simulations are designed to assess the effects of the different management options in all grid cells irrespective of actual land cover. After the spinup simulations, we thus ignore actual land-use patterns here and simulate only managed grasslands in all grid cells. The other land-use types do not matter in this analysis after the spinup, as we only consider local carbon and water dynamics to study the effects of the introduced grassland management options. We conducted a separate transient simulation run for each

of the options. For option $G_D$, additional simulations were conducted with livestock densities between 0 and 2 LSU ha$^{-1}$ in 0.2 increments.

**2.6  Analysis of model results**

**2.6.1  Classification according to average climatic conditions**

In order to relate grass growth, harvest and soil carbon to the climatic conditions under which they develop, we compute average

climate conditions. Temperature and annual precipitation per grid cell are averaged over the years 1998 to 2002 representing a medium-range period that reflects weather-related phenomena without relying on single years. Ranges of temperature and

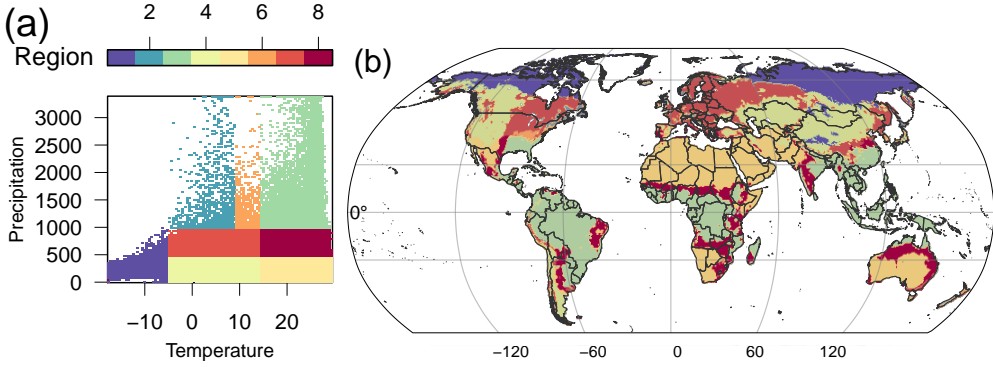

**Figure 2.** Temperature and precipitation values (a) averaged over the years 1998 to 2002. Global distribution of bioclimatic regions defined by temperature and precipitation averages (b) where each grid cell is colored depending on the bioclimatic region that it falls into.

precipitation for which grassland management results in similar changes in the carbon dynamics are classified as bioclimatic regions (Fig. 2 a) only for a better visualization of locations of similar climatic conditions (Fig. 2 b). For the analysis of the relationship between climatic conditions and grassland dynamics, we use a much finer classification, but that is difficult to visualize on a map. Under current conditions, average annual precipitation ($P_A$) is below 1000 mm a$^{-1}$ in almost all grid cells

with average annual temperatures ($T_A$) below -5 °C (bioclimatic region 1). $P_A$ values above 2000 mm a$^{-1}$ only occur with $T_A$ between -5 to 10 °C or above 15 °C corresponding to bioclimatic regions 2 and 3 (Fig. 2 a). Grid cells of bioclimatic region 2 are located along mountain ridges or near coasts whereas those of bioclimatic region 3 are located in the tropics (Fig. 2 b). High $P_A$ values mostly occur at low or at high $T_A$ but rarely with $T_A$ between 10 and 15 °C so that a kind of bimodal patterns appear (Fig.2 a for $P_A$ above 2500 mm a$^{-1}$. For the analysis in this paper, $T_A$ and $P_A$ values are divided into bins of equal

length (0.5 °C for $T_A$ and 30 mm a$^{-1}$ for $P_A$). For climate-related analyses, simulation results such as harvest or soil carbon are averaged within each bin and plotted against the $T_A$ and $P_A$ axes.

### 2.6.2   Determination of biomass use potentials under $G_D$

For management option $G_D$, we use simulations with different livestock densities to estimate the maximum annual biomass use potential for each grid cell. The livestock density under which this potential is achieved is referred to as $LSU_{harv}$. Note that

$LSU_{harv}$ is not necessarily the livestock density that can be sufficiently fed by grazing throughout the entire year, as there may be periods with insufficient grass supply (e.g. winter). The maximization of biomass use potentials under $LSU_{harv}$ livestock densities can also lead to reductions in soil carbon stocks. In order to take into account the effect on soil carbon, the carbon accumulated in the soil of each run is compared to soil carbon under option $M$. This comparison was chosen because under the mowing option neither biomass removal is maximized nor is harvested carbon added to the soil so that a rather moderate

impact on soil carbon stocks is expected compared to grassland without harvest.

To obtain the maximum livestock density that can be fed with the grass available throughout the year, the maximum livestock density is chosen under which harvest meets the demand. $LSU_{feed}$ is thus maximized with respect to the livestock that can be supported by the local grass production.

### 2.6.3 Evaluation of correspondence using Taylor diagrams

We compare observational and simulated grassland yield data by using Taylor diagrams. These allow to display three different metrics in a single diagram: the correlation coefficient of the spatial patterns, the centered Root Mean Square Deviation (RMDS) and the variance of the data sets (Taylor, 2001). As the reference data have not temporal dimension, the correlation and variance is constrained here to spatial patterns only, whereas the Taylor diagram in theory allows to assess both temporal and spatial patterns simultaneously. In Taylor diagrams, the correlation coefficient is represented by the angle, the RSMD by the distance to the location of the observational data and the variance by the distance to the origin. For details on the geometrical relationship of these three metrics see Taylor (2001). The observational data set is depicted as a point on the x-axis (perfect correlation with itself) at the value that corresponds to its variance (distance to origin of plot). Complete agreement of a simulation result with the observational data set would be expressed by the same variance as the observational data set, a RMSD of 0 and a correlation coefficient of 1.

## 3 Results

We present simulation results for the implemented management options including different livestock densities for the daily grazing option $G_D$. The effects of the management options are described for grass yield, NPP, and total soil carbon of the 3 m soil column and analyzed with respect to the underlying processes (see also discussion on strengths and weaknesses of the chosen approach in section 4.2). Simulation results are compared in different selections to a European grass harvest data set with sub-national resolution (Smit et al., 2008) and are discussed with respect to reported livestock densities. All results are presented as 5-year averages around the year 2000 (1998-2002). Variability around the mean values are presented as $\pm$ one standard deviation $(x \pm y)$.

### 3.1 Global grassland biomass

#### 3.1.1 Frequent mowing without grazing – Default option $D$

Climatic conditions are of major importance for the presented results so that we aggregate them according to temperature and precipitation values (as described in section 2.6.1). For option $D$, productivity and harvest (Fig. 3 a, b) values are high under tropical conditions ($T_A$ above $15\,^\circ$C and $P_A$ above 1000 mm, compare bioclimatic region 3 in Fig. 2). There, NPP reaches values above 800 gC m$^{-2}$ a$^{-1}$ (Fig. A1 b) and grass harvest is high ($820 \pm 90$ gC m$^{-2}$ a$^{-1}$, where the value after $\pm$ denotes the standard deviations across locations) which corresponds to 80 to 93 % of the NPP (25 and 75 % quantiles).

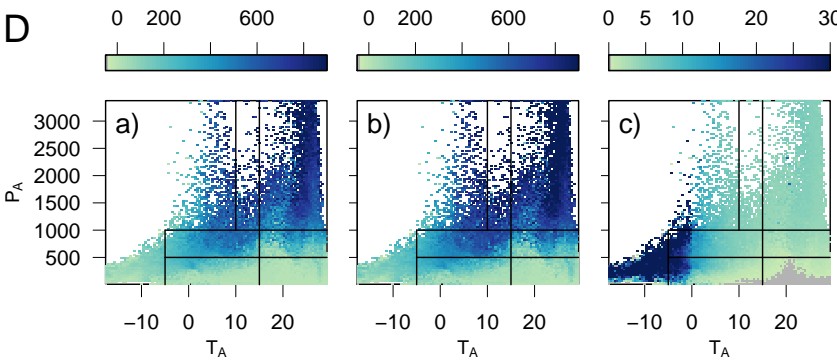

**Figure 3.** Simulation results for option $D$ averaged over the years 1998 to 2002; a) grass harvest (gC m$^{-2}$ a$^{-1}$), b) NPP (gC m$^{-2}$ a$^{-1}$), and c) soil carbon (kgC m$^{-2}$) per $T_A$ (°C) and $P_A$ (mm a$^{-1}$) values. Black lines denote bioclimatic regions as depicted in Fig. 2.

Arid and moderately warm regions ($T_A$ above 15°C and $P_A$ below 500 mm) are characterized by low NPP ($< 200$ gC m$^{-2}$ a$^{-1}$) and grass yield ($33 \pm 36$ gC m$^{-2}$ a$^{-1}$) which corresponds to 46 % of the NPP. Temperate regions ($T_A$ above -5°C and $P_A$ above 500 mm) show medium productivity (NPP between 200 and 600 gC m$^{-2}$ a$^{-1}$) and grass yield ($280 \pm 110$ gC m$^{-2}$ a$^{-1}$) so that on average 72 % of the NPP is harvested. This can be the case when the biomass increment by NPP is similar to the harvested biomass for a longer time period. When $T_A$ are on average below 0°C, harvest is relatively low ($100 \pm 49$ gC m$^{-2}$ a$^{-1}$) and about 48 to 56 % of the NPP.

The pattern for soil carbon (Fig. 3 c) is nearly inverted with low values ($5.2 \pm 0.9$ kgC m$^{-2}$) in highly productive regions. Carbon rich soils occur only in permafrost affected areas of the northern latitudes at $T_A$ below -5°C (Fig. 3 c). There, total soil carbon values exceed 30 kgC m$^{-2}$ and may reach values up to 85 kgC m$^{-2}$ although productivity is usually quite low (NPP values of $180 \pm 50$ gC m$^{-2}$ a$^{-1}$).

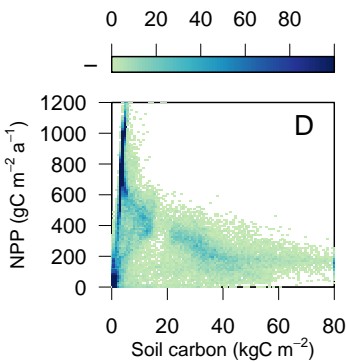

**Figure 4.** Number of grid cells with specific soil carbon (x-axis) and NPP (y-axis) value combinations. The values are based on simulation results for option $D$ averaged over the years 1998 to 2002.

In order to show which combinations of NPP and soil carbon content can occur, the same data as in Fig. 3 are presented as number of grid cells with a specific combination of NPP and soil carbon content (Fig. 4). The resulting pattern shows high values close to the origin and along a straight line from the origin with a slope of 0.24, meaning that the pathway of assimilated carbon from the atmosphere via the biosphere into the soil leads to at least 410 gC m$^{-2}$ soil carbon accumulation when NPP is about 100 gC m$^{-2}$ a$^{-1}$. Moderate to low numbers occur with higher soil carbon content only when NPP is lower (soil carbon of 20 kgC m$^{-2}$ is accompanied with NPP values between 0 and 600 gC m$^{-2}$a$^{-1}$ and this range decreases for higher soil carbon values). There are no areas with soil carbon to NPP ratios above 0.25. Likewise, high NPP and high soil carbon content do not occur together, since soil carbon contents above 30 kgC m$^{-2}$ are simulated in polar regions (compare Fig. 3), where NPP is low.

The frequent mowing option $D$ generates nearly the highest grass yield in all regions with a minimum productivity compared to other harvesting schemes. As soon as regrowth occurs, leaf biomass is removed without an additional or residual flux into the soil. Productivity under high $T_A$ and $P_A$ is even enhanced by the harvesting because of the comparatively high residuals (Fig. 1) for high leaf biomass. The feedback of leaf carbon content to photosynthesis favours plant regrowth in these regions because leaves after the harvest are growing exponentially. The moderate reduction in leaf carbon still ensures high productivity while reducing maintenance respiration so that the net increase in carbon grows overproportionally. Cold and low productive regions with low respiration and turnover provide the only environment in which high values of soil carbon are reached under option $D$.

### 3.1.2   Few mowing events without grazing – Option $M$

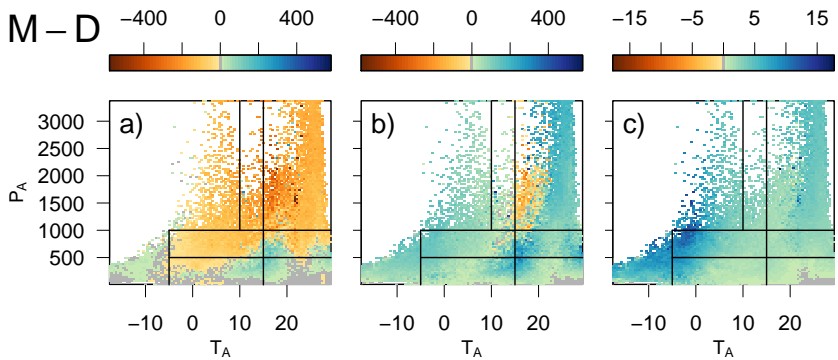

**Figure 5.** Difference of simulation results for option $M$ to those for option $D$ averaged over the years 1998 to 2002; averaged a) grass harvest (gC m$^{-2}$ a$^{-1}$), b) NPP (gC m$^{-2}$ a$^{-1}$), and c) soil carbon (kgC m$^{-2}$) per $T_A$ (°C) and $P_A$ (mm a$^{-1}$) values. Black lines denote bioclimatic regions as depicted in Fig. 2.

Management option $M$ seems to disadvantage, in terms of harvest (Fig. 5 a), areas that are highly productive under option $D$. On the other hand, it favours regions with low productivity under more frequent mowing. In particular, regions with $T_A$ above -5 °C and $P_A$ above 500 mm a$^{-1}$ show about 20 to 30 % lower harvest than for option $D$ while in less productive regions either

with $T_A$ below -5 °C in the boreal north or $P_A$ below 800 mm a$^{-1}$ in Sub-Saharan Africa, harvest amounts can be substantially increased in relative terms (up to 200 % increase).

NPP is higher for option $M$ than for option $D$ for the majority of grid cells (87 %) with an increase of more than 100 gC m$^{-2}$ a$^{-1}$ for one third of the cells. These high increases of NPP occur especially under low $T_A$ ($< 0$ °C) and $P_A$ above 500 mm a$^{-1}$ (Fig. 5 b) or higher $T_A$ (between 10 and 20 °C) and above $P_A$ of 300 mm a$^{-1}$. Only in temperate and moderately humid areas ($T_A$ between 10 and 20 °C and $P_A$ above 1000 mm a$^{-1}$), NPP values are lower than for option $D$. When NPP is low ($< 400$ gC m$^{-2}$ a$^{-1}$), grass yield is about 30 to 45 % of the NPP and this share increases with NPP to about 60 % for highly productive areas in the tropics. In these areas (e.g. East China, South-Eastern US and Southern Brazil, Fig. A2 b), harvest is also most reduced in comparison to $D$ (200 gC m$^{-2}$ a$^{-1}$).

Soil carbon for option $M$ is higher for almost all grid cells than for option $D$ ($3.5 \pm 3$ kgC m$^{-2}$). The increase is especially high in regions with $T_A$ below 0 °C and $P_A$ above 1000 mm a$^{-1}$ ($10.5 \pm 2.7$ kgC m$^{-2}$, Fig. 5 c) which corresponds to most of the boreal regions.

Also for this option, high productivity is occurring only with low soil carbon values and the combination of low productivity and high soil carbon content occurs less often (Fig. 6). The ratio between soil carbon and NPP is 0.14, i.e. 740 gC is accumulated in the soil per 100 gC annual net primary productivity. Soil carbon values above 20 kgC m$^{-2}$ are reached only for NPP values below 600 gC m$^{-2}$ a$^{-1}$ and for soil carbon contents above 40 kgC m$^{-2}$ only occur when NPP is above 75 gC m$^{-2}$ a$^{-1}$.

Under option $M$, only in unfavourable regions more biomass is harvested than under option $D$. Thus, the removal of all carbon above the threshold also under arid or cold conditions under option $M$ extracts more carbon from the plants than under the productivity-dependent harvest rule $D$. This enables even higher plant productivity and enhanced carbon sequestration in the soil. In regions with higher $T_A$ and $P_A$, the less frequent mowing does not reduce leaf carbon and therefore LAI so drastically, so that productivity increases. Reducing leaf carbon and thus LAI only affects photosynthetic capacity below an LAI of 4 and otherwise removes unproductive leaf carbon which causes higher maintenance respiration. The net effect of both

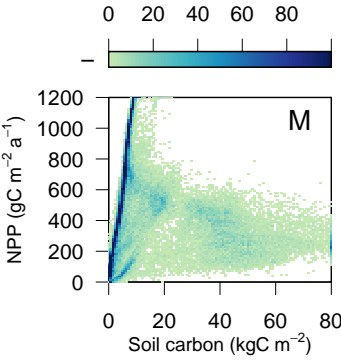

**Figure 6.** Number of grid cells with specific soil carbon (x-axis) and NPP (y-axis) value combinations. The values are based on simulation results for option $M$ averaged over the years 1998 to 2002.

harvest consequences is positive for NPP under option $M$ in comparison to option $D$ except in the above-mentioned temperate and humid areas (regions 6 and partly 3) where under option $D$ harvest and NPP were higher than 700 gC m$^{-2}$ a$^{-1}$.

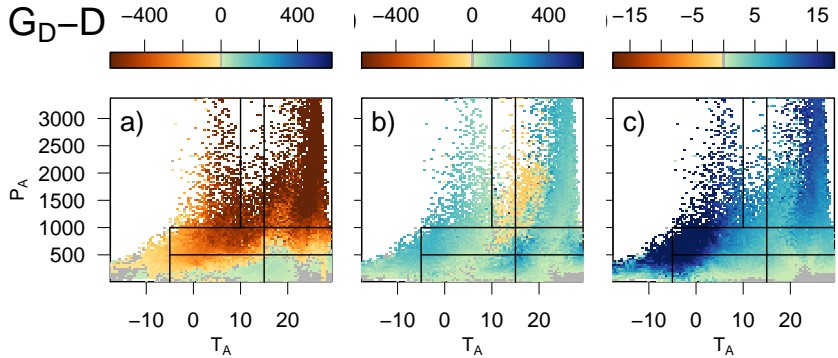

**Figure 7.** Difference of simulation results for option $G_D$ to those for option $D$ averaged over the years 1998 to 2002; averaged a) grass harvest (gC m$^{-2}$ a$^{-1}$), b) NPP (gC m$^{-2}$ a$^{-1}$), and c) soil carbon (kgC m$^{-2}$) per T$_A$ ($^\circ$C) and P$_A$ (mm a$^{-1}$) values. Black lines denote bioclimatic regions as depicted in Fig. 2.

### 3.1.3 Daily grazing without mowing – Option $G_D$

For the standard application of option $G_D$, a livestock density of 0.5 LSU ha$^{-1}$ is assumed so that grass harvest (Fig. A3 a) seems quite uniform and much lower for more productive regions than for options $D$ or $M$. This is, because the prescribed livestock density and the assumed static intake rate per day and LSU ha$^{-1}$ pre-define a maximum harvest rate that can be fulfilled in many regions. This underutilized harvest potential is quite large for tropical regions with T$_A$ above 20$^\circ$C and P$_A$ above 1000 mm a$^{-1}$ (Fig. 7 a). In drier areas with T$_A$ between 10 and 20$^\circ$C, harvest is low (48 $\pm$ 23 gC m$^{-2}$ a$^{-1}$) but about 14 gC m$^{-2}$ a$^{-1}$ higher on average than for option $D$.

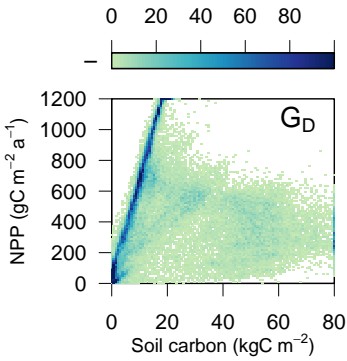

**Figure 8.** Number of grid cells with specific soil carbon (x-axis) and NPP (y-axis) value combinations. The values are based on simulation results for option $G_D$ averaged over the years 1998 to 2002.

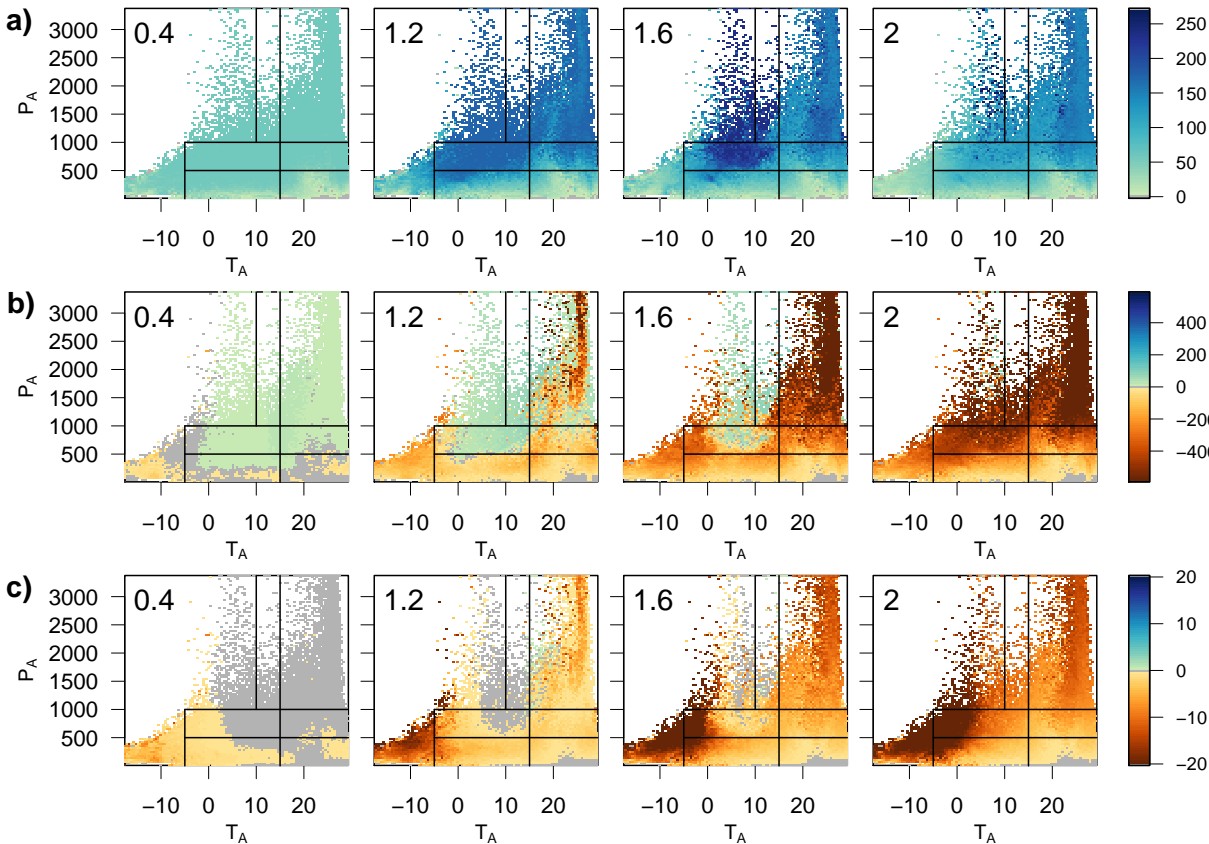

**Figure 9.** Difference of simulation results for option $G_D$ with livestock densities of 0.4, 1.2, 1.6 and 2 LSU ha$^{-1}$ to those for option $G_D$ with livestock density 0 LSU ha$^{-1}$ averaged over the years 1998 to 2002; averaged a) grass harvest (gC m$^{-2}$ a$^{-1}$), b) NPP (gC m$^{-2}$ a$^{-1}$), and c) soil carbon (kgC m$^{-2}$) per T$_A$ (°C) and P$_A$ (mm a$^{-1}$) values. Black lines denote bioclimatic regions as depicted in Fig. 2.

The difference in harvest has implications on the productivity itself. NPP is mostly increased in comparison to option $D$ by 8 to 60 % (Fig. A3, b). Although harvest is lower for all regions with P$_A$ above 500 mm a$^{-1}$ (Fig. 7 a), the average NPP is also reduced in regions with T$_A$ between 10 and 20 °C (Fig. 7 b). This is the case in the south of China and in the south of Brazil (Fig. A3 b).

5      Soil carbon for option $G_D$ is on average twice as high as for option $D$ (20.3 ± 20 kgC m$^{-2}$ Fig. A3 c, in comparison to 11.9 ± 16.9 kgC m$^{-2}$, Fig. A1 c) since this option includes a flux of the harvested carbon (grazed biomass) into the soil in form of manure. When decomposition rates are low (T$_A$ below 0 °C), soil carbon increases by 13.2 ± 8.4 kgC m$^{-2}$ (Fig. 7 c) in comparison to option $D$.

     The homogeneous carbon removal in option $G_D$ is rather low for most of the land area so that the productivity is mostly 10   less influenced by harvest than in option $D$. For most areas, this is stimulating for the productivity. Interestingly, in temperate regions with medium P$_A$ (between 1500 and 2000 mm a$^{-1}$) the reduction of grass harvest also coincides with less NPP. Harvest

under option $D$ was rather high (700-800 gC m$^{-2}$ a$^{-1}$) and NPP as well (800-900 gC m$^{-2}$ a$^{-1}$) so that we can state that grass harvest in these areas keeps leaf carbon in the value range for exponential growth. The soil carbon difference to option $D$ in these areas is rather small but positive because of the additional flux of manure carbon into the soil.

For daily grazing, the relationship of productivity and soil carbon content shows an even lower slope than for option $M$ (Fig. 8). The ratio between soil carbon and NPP is 0.063, so that 1590 gC is accumulated in the soil per 100 gC annual net primary productivity. Soil carbon values above 40 kgC m$^{-2}$ are reached for NPP values below 700 gC m$^{-2}$ a$^{-1}$ and for soil carbon contents above 60 kgC m$^{-2}$ only occur when NPP is above 100 gC m$^{-2}$ a$^{-1}$. In comparison to the mowing options $D$ and $M$, the loss of carbon from the system by harvest is reduced so that the transfer of NPP into the soil leads to a higher accumulation in the soil per NPP. In the case of a livestock density of 0 LSU ha$^{-1}$, i.e. neither export of carbon via harvest nor additional transfer into the soil via manure, the accumulation results in 1560 gC per 100 gC NPP.

Variation of the livestock density has a distinct effect on carbon stocks and fluxes. With increasing density from 0 to 2 LSU ha$^{-1}$, the harvested biomass (Fig. 9 a) first increases (48 $\pm$ 19 gC m$^{-2}$ a$^{-1}$ at 0.4 LSU ha$^{-1}$ until 111 $\pm$ 65 gC m$^{-2}$ a$^{-1}$ at 1.2 LSU ha$^{-1}$) and then only increases further in regions with T$_A$ between 0 and 10$^\circ$C and P$_A$ above 1000 mm a$^{-1}$ (from 173 $\pm$ 16 gC m$^{-2}$ a$^{-1}$ at 1.2 LSU ha$^{-1}$ to 208 $\pm$ 51 gC m$^{-2}$ a$^{-1}$ at 1.6 LSU ha$^{-1}$). Under even higher grazing pressure, yields decrease everywhere to an average of 79 $\pm$ 56 gC m$^{-2}$ a$^{-1}$ at 2 LSU ha$^{-1}$.

Increasing livestock densities also affect NPP in both directions (Fig. 9 b). Areas with medium-range T$_A$ (between 0 and 10$^\circ$C) and sufficient P$_A$ (above 1000 mm a$^{-1}$) are quite productive without grazing animals (736 $\pm$ 140 gC m$^{-2}$ a$^{-1}$) and even moderately increase in NPP until 1.2 LSU ha$^{-1}$ (gain of 35 $\pm$ 36 gC m$^{-2}$ a$^{-1}$). For tropical regions with T$_A$ above 20$^\circ$C and P$_A$ above 1500 mm a$^{-1}$, the productivity is negatively affected above 1.0 LSU ha$^{-1}$ (on average 260 gC m$^{-2}$ a$^{-1}$ less for 1.2 LSU ha$^{-1}$ until 840 gC m$^{-2}$ a$^{-1}$ for 2 LSU ha$^{-1}$).

Soil carbon is decreasing with increasing livestock density (globally from 21.3 $\pm$ 22.8 kgC m$^{-2}$ at 0 LSU ha$^{-1}$ to 10.8 $\pm$ 15.5 kgC m$^{-2}$ at 2 LSU ha$^{-1}$). Particularly, strong declines in soil carbon with stocking densities are simulated in cold regions

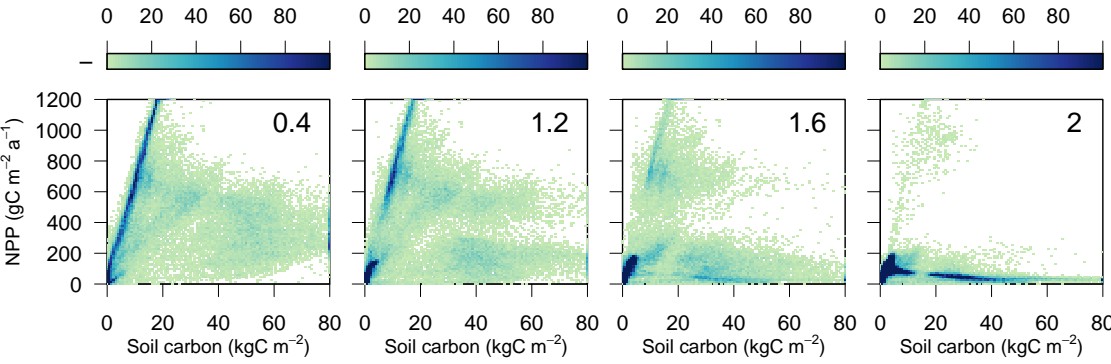

**Figure 10.** Number of grid cells with specific soil carbon (x-axis) and NPP (y-axis) value combinations. The values are based on simulation results for option $G_D$ averaged over the years 1998 to 2002. Simulation results for option $G_D$ with livestock densities of 0.4, 1.2, 1.6 and 2 LSU ha$^{-1}$ averaged over the years 1998 to 2002.

($T_A$ below $0\,^\circ$C) (Fig. 9 c) where also soil carbon stocks are large enough to allow for such strong reductions. There, soil carbon for low stocking densities (0.4 LSU ha$^{-1}$) decreases by $2.5 \pm 2.8$ kgC m$^{-2}$ and for higher densities (2 LSU ha$^{-1}$) even by $19.9 \pm 8.2$ kgC m$^{-2}$. In regions where NPP is mostly affected by increasing livestock densities ($T_A$ above $20\,^\circ$C and $P_A$ above 1500 mm a$^{-1}$), soil carbon losses are less strong ($11.8 \pm 2.9$ kgC m$^{-2}$ at 2 LSU ha$^{-1}$). Interestingly, soil carbon is not or only marginally reduced ($0.5 \pm 1.4$ kgC m$^{-2}$ for 1.2 LSU ha$^{-1}$ and $3.9 \pm 6.4$ kgC m$^{-2}$ for 1.6 LSU ha$^{-1}$) for medium-range $T_A$ (0 to $10\,^\circ$C) and sufficient $P_A$ (above 1000 mm a$^{-1}$) in which NPP increased under livestock densities of up to 1.6 LSU ha$^{-1}$.

When conditions are neither too cold nor too arid (all regions except 1, 4 and 5), a stocking density of 0.4 LSU ha$^{-1}$ results in low harvest values but stimulated NPP and unchanged soil carbon in comparison to non-grazed pasture (left panels in Fig. 9). Thus, the flux of harvested grass into the soil via manure equals on average the accumulation rates without harvest. Only in the region susceptible to a positive effect of harvest on NPP (compare Fig. 7 b), soil carbon values even increase under stocking densities between 1.2 and 1.6 LSU ha$^{-1}$. Here, the positive feedback of leaf carbon reduction on photosynthesis is outweighing the loss of carbon through grazing.

Increasing livestock densities have several effects on the productivity (Fig. 10). On the one hand, overall NPP values decrease, but the relation between soil carbon and NPP shows very slithly increasing slopes (from 0.065 for 0.4 LSU ha$^{-1}$ to 0.072 for 2.0 LSU ha$^{-1}$). Under slightly higher grazing pressure (1.2 LSU ha$^{-1}$), cells with NPP values between 500 and 700 gC m$^{-2}$ a$^{-1}$ are rather unchanged whereas those who showed NPP below 500 gC m$^{-2}$ a$^{-1}$ for lower livestock densities now have lower NPP and soil carbon content. With high grazing pressure, less and less grid cells maintain high NPP and more and more areas are characterized with NPP below 200 gC m$^{-2}$ a$^{-1}$ and soil carbon content below 40 kgC m$^{-2}$. Thus, we can show how increasing grazing pressure deteriorates NPP and soil carbon content.

### 3.1.4 Rotational grazing without mowing – Option $G_R$

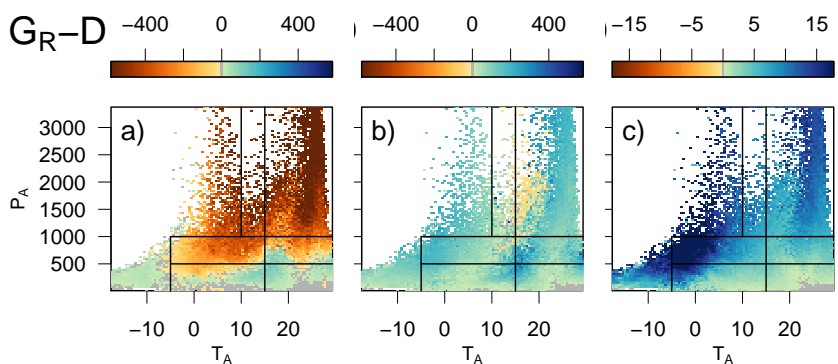

**Figure 11.** Difference of simulation results for option $G_R$ to those for option $D$ averaged over the years 1998 to 2002; averaged a) grass harvest (gC m$^{-2}$ a$^{-1}$), b) NPP (gC m$^{-2}$ a$^{-1}$), and c) soil carbon (kgC m$^{-2}$) per $T_A$ ($^\circ$C) and $P_A$ (mm a$^{-1}$) values. Black lines denote bioclimatic regions as depicted in Fig. 2.

Option $G_R$ is usually used with a livestock density of 1.2 LSU ha$^{-1}$, i.e., more than a twofold livestock density than assumed for $G_D$. Thus, harvest values in productive areas are higher than $G_D$ with maximum values of 170 gC m$^{-2}$ a$^{-1}$ (Fig. A4 a) but with very similar NPP (Fig. A4 b) as in $G_D$ (Fig. A3 b). Grass harvest shows similar spatial patterns for both grazing options (compare Fig. A4 a and A3 a). Nevertheless, for option $G_R$ the required demand of grass harvest for the given livestock density

is met on 42 % of the pasture area whereas the demand is met at 67 % of the area for option $G_D$ (19). Under favorable growth conditions with T$_A$ above 20 °C and P$_A$ above 1000 mm a$^{-1}$, harvest is much lower than for option $D$ (Fig. 11 a) but the given demand can be fulfilled for all grid cells. In drier areas with T$_A$ between 10 and 20 °C, harvest is low (78 $\pm$ 58 gC m$^{-2}$ a$^{-1}$) but about 44 gC m$^{-2}$ a$^{-1}$ higher on average than for option $D$.

The effect of harvest on productivity is similar as for option $G_D$. NPP is mostly increased in comparison to option $D$ namely

by 24 to 138 gC m$^{-2}$ a$^{-1}$ or 11 to 63 % (Fig. A4 b). As for option $G_D$, harvest is lower for all regions with P$_A$ above 500 mm a$^{-1}$ (Fig. 11 a) and average NPP is reduced in regions with T$_A$ between 10 and 20 °C (Fig. 11 b).

Soil carbon for option $G_R$ is on average a bit lower than for option $G_D$ (19.8 $\pm$ 21 kgC m$^{-2}$, Fig. A4 c) because this option prescribes a higher livestock density but it results in similarly higher soil carbon content in comparison to option $D$ because of the additional carbon flux via manure application. The soil carbon increase in regions with low T$_A$ (below 0 °C) is with

11.4 $\pm$ 6.4 kgC m$^{-2}$ (Fig. 11 c) a bit lower than for option $G_D$ (compare Fig. 7 c).

For rotational grazing, the occurrence of productivity and soil carbon content is rather similar to option $G_D$ (Fig. 12). The relation between soil carbon and NPP shows a similar slope of 0.065 and also the occurrence of higher soil carbon values goes along with NPP values between 700 and 100 gC m$^{-2}$ a$^{-1}$.

The general pattern for this option compares to the daily grazing option $G_D$ under higher livestock densities (Fig. A3) with

lesser differences to the default option (compare lower values in Fig. 11 than in Fig. 7). Both grazing options give diverging results when climatic conditions show strong seasonality. There, the intermittent intensive grazing with longer recovery periods better exploits local productivity than the continous carbon removal at the same livestock density without recovery periods.

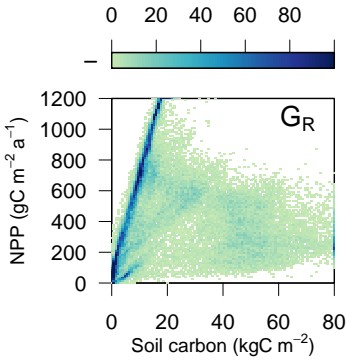

**Figure 12.** Number of grid cells with specific soil carbon (x-axis) and NPP (y-axis) value combinations. The values are based on simulation results for option $G_R$ averaged over the years 1998 to 2002.

### 3.2 Comparison to observed harvest data

We use estimates of grassland harvest rates from Smit et al. (2008) to evaluate simulated grass harvest at the European scale. These data were provided for European administrative units (NUTS-2 level) or country-level, when more detailed data were not available (Fig. 13 a). We compare these data to our simulation results in three different combinations (see specifications below). Since management assumptions for the simulations were spatially homogeneous and management in Europe is known to vary spatially as well as temporally, we combine different simulated management options (one option per grid cell) to best match the observed productivity to find out whether climate- and management-induced variations in grass harvest can be captured by the applied options. Therefore, we formulate and test the hypotheses that observed European grass harvest

1. levels can be reached by grazing animals only,

2. patterns can be reproduced by varying stocking densities under option $G_D$ and

3. patterns can be reproduced by combining the management options described here.

For testing hypothesis 1, we choose for each regional value from the simulation results for option $G_D$ with varying livestock densities the maximum harvest value (see section 2.6.2, Fig. 13 b) which resulted mostly from simulations with medium stocking densities. Clearly, the pattern differs from the reported yield estimates (Fig. 13 a). The gradient of reported yields from northeast to southwest is underestimated and yields are higher in southern Europe and lower in the western parts of the continent. Thus, a continuous withdrawal of leaf biomass could achieve higher grass harvest in eastern Europe and the Mediterranean whereas for yields in western Europe (esp. Great Britain, The Netherlands and Norway) much higher values are reported than simulated. Therefore, we can reject hypothesis 1 and, thus, support the assumption that grassland management in Europe is not homogeneous concerning the presence of animals on the pasture or the harvesting intensity.

Hypothesis 2 is assessed by testing whether the reported pattern with the pronounced gradient from northwest to southeast can be reproduced. From simulations with option $G_D$, we select per region the harvests closest to the reported values (Fig. 13 c) so that the livestock densities can be inferred that lead to the observed harvest values. The resulting pattern matches the reported values below 260 gC m$^{-2}$ a$^{-1}$ which are occurring in most of southern and eastern Europe as well as Scandinavia apart from Norway. Values are only underestimated in Great Britain, The Netherlands, Ireland and Norway on highly managed grasslands, e.g. which are fertilized and irrigated. Comparing those regions in which maximum values (Fig. 13 b) are the closest to reported values (Fig. 13 a) but still more than 10 % too low, apart from the 4 countries mentioned before, German and French provinces appear. We interpret this as strong indicators for intensively managed systems. On the other hand, regions in which maximum harvest by grazing overestimates reported values by 50 % are located mainly in East European countries (Slovakia, Montenegro, Macedonia, Lithuania, Hungary, Croatia, Estonia, Belarus, Bulgaria, Albania). There, the potential of grass production is by far not utilized while climatic conditions are not limiting. Hypothesis 2 can be confirmed by these findings.

For testing hypothesis 3, for each geographical region the closest value from all available simulation results is chosen (Fig. 13 d). The derived values deviate from those in Fig. 13 c in the highly managed countries identified before (Great Britain,

The Netherlands and Ireland) and some regions in Finland, Germany and France. The options that result in values more similar to observations are the default option $D$ and mowing $M$ both describing pasture regimes with different (more in case of $D$ and less frequent in case of $M$) harvest frequencies that increase yields. Regions in which reported yields are higher than in Fig. 13 d can thus be identified as regions in which the model cannot reproduce the observed productivity levels. Here other yield increasing measures such as fertilization and irrigation may play a role that are not considered in the model. Therefore, also hypothesis 3 can be confirmed by analysing the simulation results.

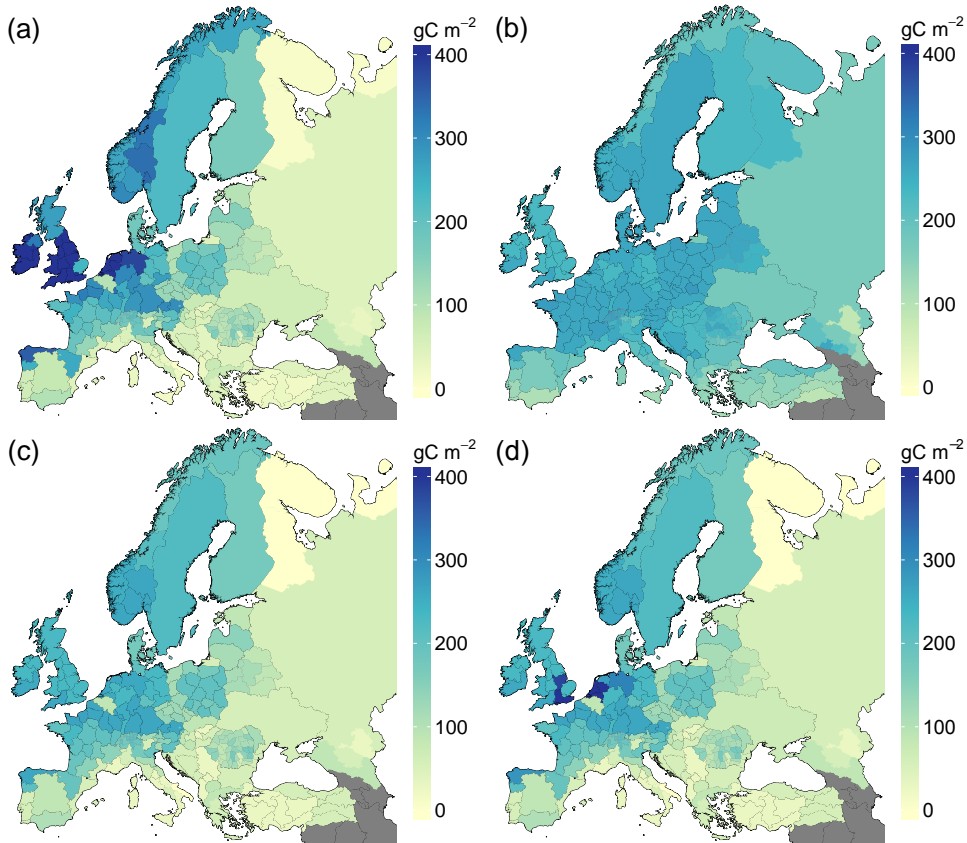

**Figure 13.** Average grassland harvest for European geographical regions (in gC m$^{-2}$ a$^{-1}$); (a) as given by Smit et al. (2008), (b) maximum grass harvest as simulated by option $G_D$ with varying livestock densities, (c) simulated grass harvest as simulated with option $G_D$ with the livestock density that produces harvest values closest to observed values and (d) simulated grass harvest as simulated all options that produces harvest values closest to observed values.

Results of all presented simulation combinations in Fig. 13 are compared to the estimates by Smit et al. (2008) within the Taylor diagram (Fig. 14) represented by different symbols. Application of option G$_D$ and selecting maximum values from simulations with varying livestock densities (Fig. 13 b) does underestimate the variability of the reported values and has a correlation coefficient of 0.44 (Fig. 14, orange dot). Selecting those livestock densities for option $G_D$ which result in harvest

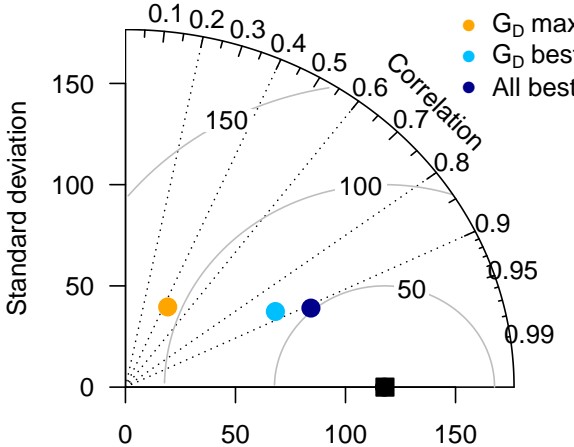

**Figure 14.** Taylor diagram of simulated European grass harvest (gC m$^{-2}$ a$^{-1}$) aggregated to administrative units and evaluated using estimates from Smit et al. (2008) for combinations of grassland management options. 'G$_D$ max' denotes values diplayed in Fig. 13 b, 'G$_D$ best' in Fig. 13 c, 'All best' in Fig. 13 d. Perfect match to the estimates would mean that the simulated data points (colored dots) are at the location of the black box on the x-axis, having a standard deviation of 118 gC m$^{-2}$ a$^{-1}$ (distance to origin) and a correlation of 1 (angle) as well as a centered RMSD of zero (gray contour lines).

values closest to reported values (Fig. 13 c) does increase the standard deviation from 44 to 78 gC m$^{-2}$ and the correlation coefficient to 0.88 (Fig. 14, lightblue dot). Including all simulation results also with mowing and the default harvest options (Fig. 13 d) results in a standard deviation of 93 gC m$^{-2}$ and a correlation coefficient of 0.91 (Fig. 14, blue dot). The diversity of management options and stocking densities as implemented in LPJmL thus helps to reproduce the spatial variability in
European grass yields, but fails to capture the high productive areas, leading to an underestimation of the variance (standard deviation of observed yields is 118 gC m$^{-2}$ a$^{-1}$).

### 3.3   Sustainable potentials

In order to derive biomass use potentials for livestock grazing at the global scale, simulation results with option $G_D$ and livestock densities between 0 and 2 LSU ha$^{-1}$ were analyzed (see section 2.6.2). First, the livestock density value LSU$_{harv}$
was determined for each grid cell for which the maximum harvest was simulated (Fig. 15a). High values can be found in temperate regions in Europe and the US mainly corresponding to bioclimatic region 7 (see Fig. 2). Arid regions (P$_A$ below 500 mm a$^{-1}$) show low values but large livestock densities are feasible e.g. in Australia or in northern Russia. This is the case for regions e.g. with strong seasonality in T$_A$ or P$_A$ and a distinct but short vegetation period. There, the detected LSU$_{harv}$ results in the highest yield but livestock at this stocking density cannot be sustained throughout the year by the pasture area, because the daily feed demand cannot be met during the less productive periods. Also, negative impacts, such as possible
reductions in NPP or soil carbon, are neglected when determining LSU$_{harv}$ only for maximizing the yield.

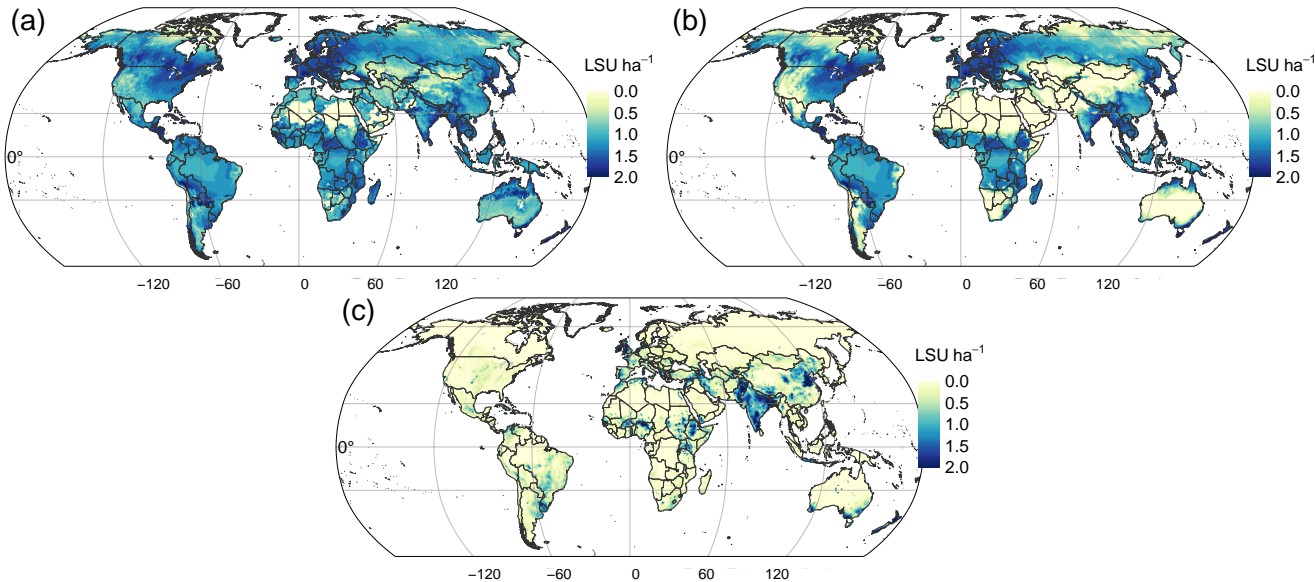

**Figure 15.** Distribution of livestock densities that result in (a) maximum harvest ($LSU_{harv}$ in LSU ha$^{-1}$) and (b) maximum LSU that can be continuously supported by grazing only ($LSU_{feed}$ in LSU ha$^{-1}$) under harvest option $G_D$ averaged over the years 1998 to 2002. Reported livestock densities in pastoral and mixed livestock production systems (Robinson et al., 2014) are given as comparison in c).

The distribution of the maximum livestock density that can be fully supported by the local grass production ($LSU_{feed}$) (Fig. 15b) differs from $LSU_{harv}$ especially in arid regions like inner Australia, North Africa, western North America and the Middle East where values close to 0 LSU ha$^{-1}$ are derived for $LSU_{feed}$. Also in polar areas in North America and Asia (region 1 in Fig. 2), values for $LSU_{feed}$ are reduced considerably compared to $LSU_{harv}$. Even though grass productivity can be high

in parts of the year, this is not sufficient to continuously supply the feed demand of higher livestock densities.

Comparing both potentials ($LSU_{harv}$ and $LSU_{feed}$) with reported livestock densities by Robinson et al. (2014) (Fig. 15c), one has to consider that livestock can only be reported on actual grassland area. In areas where livestock is reported, values of the potential $LSU_{feed}$ are mostly higher, but in arid regions (e.g. North Africa, Northern China, Middle East countries or South Africa) the sustainable potentials derived from LPJmL are lower than reported values. When comparing the LSU reported by

10 Robinson et al. (2014) to $LSU_{harv}$, LPJmL derived values are higher than reported except for a few regions (e.g. Northern China and in the North of India) where feed baskets for ruminant livestock contain only minor shares of grass (Herrero et al., 2013). This is consistent with the expectation that potentials are higher than reported values.

The global distribution of $LSU_{harv}$ gives the highest share of more than 20 % of the grid cells for 1.2 LSU ha$^{-1}$ (Fig. 16). $LSU_{harv}$ is between 1 and 1.4 LSU ha$^{-1}$ for 56 % of the cells. The harvest achieved in grid cells with $LSU_{harv} = 1.2$ LSU ha$^{-1}$

is 151 $\pm$ 43 gC m$^{-2}$ a$^{-1}$, which is well below the actual demand of 175 gC m$^{-2}$ a$^{-1}$ under this livestock density. The gap between demand and simulated yield is even higher for low productive regions. In 6 % of the cells, no livestock can be fed by grazing, and in the 2.5 and 3.7 % for which 0.2 or 0.4 LSU ha$^{-1}$ are derived as $LSU_{harv}$, average yield is 11.3 and

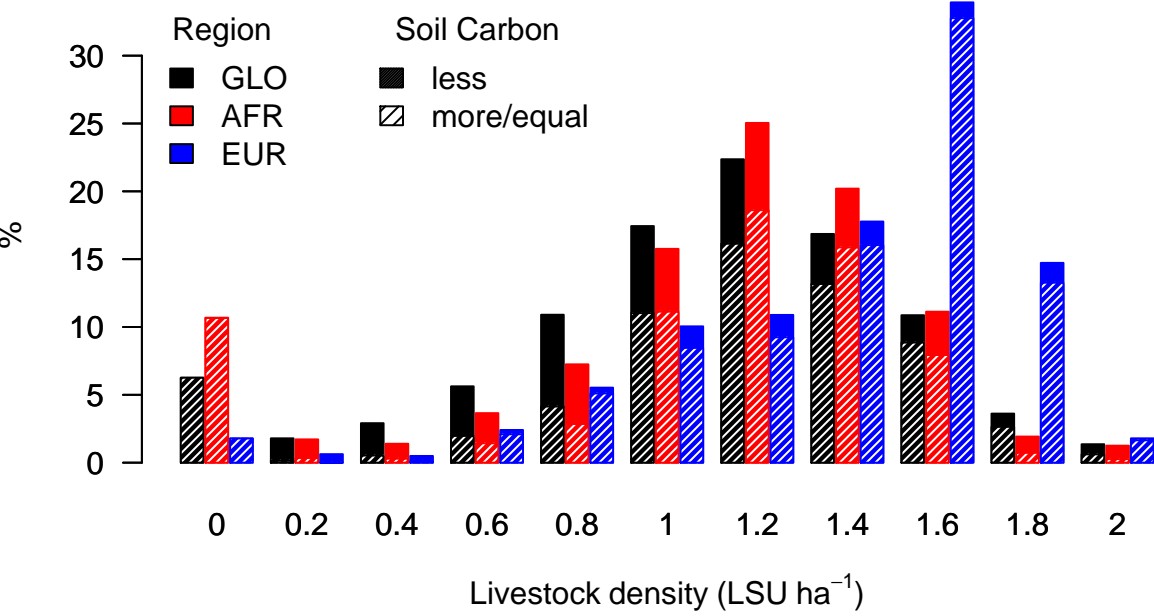

**Figure 16.** Distribution of livestock densities under which harvest with option $G_D$ is maximal ($\mathrm{LSU}_{harv}$, section 2.6.2) as percentage of the area in the respective regions. Colors denote global values (GLO) as well as regional (EUR for Europe and AFR for Sub-Saharan Africa). The hatched sections of the bars depict the area in which maximum harvest is achieved without soil carbon loss in comparison to results under the mowing option $M$.

33 gC m$^{-2}$ a$^{-1}$ instead of the 29.2 or 58.4 gC m$^{-2}$ a$^{-1}$ required by the grazing animals. These regions – belonging to rather arid or cold climates (Fig. 15) – are suitable for some grass production but only during a short vegetation period. This can be underlined when deriving the distribution of $\mathrm{LSU}_{harv}$ only for Sub-Saharan Africa (AFR) or Europe (EUR) separately (red and blue bars in Fig. 16). In AFR, about 11 % of the grid cells cannot feed any livestock and $\mathrm{LSU}_{harv}$ livestock densities above

5   1.2 LSU ha$^{-1}$ are found for 34 % of the cells which is similar to the global pattern (32 %) but much lower than European distributions of $\mathrm{LSU}_{harv}$ (67 %). In grid cells with $\mathrm{LSU}_{harv}$ of 1.8 LSU ha$^{-1}$, grass harvest is globally 232 $\pm$ 72 gC m$^{-2}$ a$^{-1}$, only 135 $\pm$ 102 gC m$^{-2}$ a$^{-1}$ for AFR and 261 $\pm$ 12 gC m$^{-2}$ a$^{-1}$ for EUR. For one third of the European grid cells (32.5 %) $\mathrm{LSU}_{harv}$ is calculated as 1.6 LSU ha$^{-1}$ with grass harvest of 233 $\pm$ 7 gC m$^{-2}$ a$^{-1}$ which is very close to the given demand of 234 gC m$^{-2}$ a$^{-1}$ at this livestock density.

10   In the assessment of sustainable production potentials, we also account for negative impacts of grassland management, by considering the soil carbon losses. For each of the grid cells, the soil carbon content under the derived $\mathrm{LSU}_{harv}$ is compared to soil carbon for option $M$ (see section 2.6.2). The bars in Fig. 16 are hatched for those grid cells, in which the difference of both soil carbon values is positive, i.e. there is no additional loss of soil carbon compared to option $M$. Globally, only 65 % of the

area falls into this category, whereas this is the case for 70 % in AFR and 91 % in EUR. Especially in low productive regions with $LSU_{harv}$ values below 1 LSU ha$^{-1}$, most of the grid cells have lower soil carbon content under $G_D$ than under $M$ when the $LSU_{harv}$ livestock density is chosen (e.g. 91, 82 or 65 % of the grid cells with $LSU_{harv}$ of 0.2, 0.4 or 0.6 LSU ha$^{-1}$). In Argentina, 48 % have $LSU_{harv}$ values between 0.6 and 1 LSU ha$^{-1}$ and 58 % of these areas are subject to soil carbon loss

in comparison to the simulation with option $M$. Although the difference in soil carbon is negative for low $LSU_{harv}$ also in Sub-Saharan Africa and Europe (80 % in AFR and 98 % in EUR for $LSU_{harv} = 0.2$ LSU ha$^{-1}$), this relation is quite different for higher $LSU_{harv}$. For the cells with a $LSU_{harv}$ of 1 LSU ha$^{-1}$, one third in AFR and only 16 % in EUR show lower soil carbon content than under option $M$. 11 % of the area in AFR have maximum harvest at livestock densities of 1.6 LSU ha$^{-1}$ and 7.9 % of the African area sees no carbon loss in comparison to option $M$ in this $LSU_{harv}$ category. For EUR, these

fractions are much larger in this category with 34 % and 33 % respectively. Thus, the potential of high grassland yields and high livestock densities in Europe without diminishing soil carbon content is much higher than for most parts of the world.

## 4 Discussion

### 4.1 Major findings & model performance

We find that the representation of the feedbacks between primary productivity, respiration, carbon turnover and soil dynamics
as implemented in LPJmL allows for simulating the effects of grass harvesting options as reported in the literature.

Grassland management has a strong impact on the carbon cycle, i.e. it alters yields and productivity (NPP) as well as carbon stocks in vegetation and soil. We find that increasing densities of grazing livestock can lead to both positive and negative changes in NPP and soil carbon content depending on the climatic conditions, which is in agreement with various field studies: Negative impacts of overgrazing on both productivity and soil carbon are reported in many studies (e.g. Schuman et al., 1999;
Reeder and Schuman, 2002). The potential to re-establish soil sequestration by grazing management (reduction of herd sizes or exclusion of livestock) was reported from a study in China (Wang et al., 2011).

Especially in temperate and humid regions, simulated yield, productivity as well as soil carbon can increase even under moderate to high grazing livestock densities (Fig. 9). In temperate regions in the US (Conant et al., 2001) as well as in Europe (Soussana et al., 2004, 2010), carbon sequestration was measured for moderate grazing and soil improving techniques such
as moderate fertilization. In Europe, the soil carbon sequestration potential was estimated to be higher for cut than for grazed grassland from a 2-year study (Soussana et al., 2007, 2010) which we could not reproduce with the current mowing option $M$, but assume that this is due to the simplifying assumptions made (no variation in timing and number of harvest events). In boreal regions, simulated soil carbon reduction with increasing grazing pressure is stronger than the NPP decline. Here, carbon rich soils with more than 60 kgC m$^{-2}$ develop only without grazing animals under low NPP and respiration.
In arid regions of the US, productivity and soil carbon content increased under light grazing (Schuman et al., 1999; Reeder and Schuman, 2002) due to an acceleration of carbon turnover. We could reproduce these dynamics by simulations with option $G_D$ and livestock densities below 0.4 LSU ha$^{-1}$ (compare bioclimatic region 4 in Fig. 9). Even a more diverse plant community and denser rooting system were observed under these conditions (LeCain et al., 2002; Reeder and Schuman, 2002).

In tropical regions, grazing even with low livestock densities was observed to decrease soil carbon in the Amazon (Fearnside and Barbosa, 1998) and in Argentina (Abril and Bucher, 1999) when soil management did not include additional fertilization. In comparison to temperate regions, simulated carbon stocks under option $G_D$ are much lower in tropical regions ($< 10\,\mathrm{kgC\,m^{-2}}$) at low livestock densities and show a much higher reduction in NPP under increasing stocking densities (Fig. 9b).

The newly implemented options therefore allow for an assessment of different pasture management options as well as for accounting for these important dynamics in global simulations of biogeochemical cycles or agricultural productivity. In comparison to the European grassland productivity (Smit et al., 2008), the presented results only reproduce the observed values and variability when non-homogeneous livestock densities and mowing schemes are considered. This supports the relevance of grassland management for bio-geochemical and agricultural analyses. They can be applied for evaluating potentials for sustainable intensification or deriving suitable pasture areas for livestock production under current and future climatic conditions.

## 4.2 Uncertainties & assumptions

Implementing grassland management options into a DGVM requires some assumptions and parameter value choices. We discuss these in comparison to other approaches.

Mowing is a harvest technique applied mostly in Europe on pasture and productive grassland often in combination with grazing before or after the mowing event. We assume two cuts per year and the removal of all leaf carbon above a threshold of $25\,\mathrm{gC\,m^{-2}}$. This simplifying approach is easily applied and captures the difference to productivity-dependent harvesting (default option $D$) and the grazing regimes. The choice of the cutting dates as $1^{st}$ of June and $1^{st}$ of December takes into account a spring and winter cut for both hemispheres. For simplicity we stick to this regular scheme and only allow for skipping mowing events if not enough leaf biomass is available. For Germany, phenological data are collected by the German Weather Service (Kaspar et al., 2014) also for haymaking dates which give beginning of June for the first cut (average day of year 157 $\pm$ 12 days). Usually, the spring cut is triggered by a threshold amount of biomass build up before the flowering. This threshold is given e.g. as $1700\,\mathrm{kg\,DM\,ha^{-1}}$ ($= 77\,\mathrm{gC\,m^{-2}}$) in recommendations for pastures in temperate regions (IKC, 1993) but it is hardly possible to derive a common rule for a global application from literature values. Also information on conditions for a second mowing is scarce with estimates such as leaf biomass accumulation of $3000\,\mathrm{kg\,DM\,ha^{-1}}$ ($= 135\,\mathrm{gC\,m^{-2}}$) or day of year 222 ($\pm$ 20 days) in the German phenological database. For the reduction of frost damage (IKC, 1993), it is recommended to mow not too late in fall (e.g. after the $1^{st}$ of October in the northern hemisphere). Thus, our choice for a spring cut in the northern and southern hemisphere and an additional cut where enough leaf biomass is present represent the affirmed practices for mowing. The poor timing of the winter cut, which is typically simulated too late in comparison to observational data (Kaspar et al., 2014) is acceptable as long as grass quality is not considered, as leaf carbon does not change much during the cold period of the year (very low photosynthesis and maintenance respiration). The biomass residual of $25\,\mathrm{gC\,m^{-2}}$ is lower than the parameter chosen for the ORCHIDEE model ($150\,\mathrm{g\,DM\,m^{-2}} = 67.5\,\mathrm{gC\,m^{-2}}$, Chang et al., 2013).

Although differences in livestock grazing exist between systems globally, the major differences to other treatments can be expressed by the chosen options $G_D$ and $G_R$ and the ability to vary livestock densities. Livestock is either reared in-

house and gets additional feed from cropland or is present on the pasture daily or for a short time period. Here, the focus lies on the grassland treatment and not directly on the development and feed security of the livestock itself. This can be addressed with the presented modeling approach with the stocking density specifying the maximum removal of leaf carbon and the grazing period specifying the timing and duration of leaf carbon removal. The threshold for the beginning of grazing
is chosen differently for the continuous grazing with 5 gC m$^{-2}$ and rotational grazing with 40 gC m$^{-2}$ which represents management on pasture of different productivity. For rotational grazing, some estimates can be found in recommendations by IKC (1993) with about 700 kg DM ha$^{-1}$ (= 30 gC m$^{-2}$) or Blanchet et al. (2003) with a minimum of 4 inches of grass with 250 lb DM acre$^{-1}$ inch$^{-1}$ (= 40 gC m$^{-2}$). Williams and Hall (1994) do not give a minimum but optimal grass height of 7 inches and 300 lb DM acre$^{-1}$ inch$^{-1}$ (= 84 gC m$^{-2}$) and Undersander et al. (2002) recommend the beginning of the grazing
period at 4 to 8 inches of grass (= 40 to 48 gC m$^{-2}$). The end of the grazing period with 5 gC m$^{-2}$ is chosen a bit lower than other modeling studies (e.g. for the ORCHIDEE model: 300 kg DM ha$^{-1}$ = 13.5 gC m$^{-2}$, Chang et al., 2013).

Assumptions are necessary also with respect to the specification of the spatial distribution of management activities (Chang et al., 2013; Sandor et al., 2017). Here, we chose to analyze scenarios with spatially homogeneous management variations. With the current uncertainties in livestock density distributions (Kruska et al., 2003), management intensity and even distribution
of pasture areas (Portmann et al., 2010) a global assessment of the current role of grassland management for biogeochemical cycles is not feasible. Recently, a global dataset of grassland management was derived by combining regional data on livestock and its feed demand with satellite data and model simulations (Chang et al., 2016). Although this dataset might be of great importance for improving managed grassland assessments, Chang et al. (2016) also state that their method results in biomass production deficits in some areas caused by uncertainties in production system settings, pasture area distributions and in the
mapping of regional averages to gridded data. Thus, further research and harmonization of these approaches are needed.

## 4.3   Further development

The implementation of the four basic harvesting schemes on managed grassland is a starting point for a variety of possible further developments, applications and extensions. In order to reflect the actual contribution of managed grasslands to global biogeochemical fluxes, improvements of global datasets (such as Chang et al., 2016) on livestock densities and management
settings are needed. The intensity of the grassland management with fertilization and irrigation could be included as well corresponding to the intensity level that was established in Fader et al. (2010). This would enable to better capture highly productive regions in the United Kingdom or the Netherlands and to assess pathways to conventional or sustainable intensification management. It also seems sensible to establish combined harvesting schemes with cutting and grazing events in LPJmL when appropriate information about the application of such schemes is available. A promising approach to extract cutting events
on grassland from remote sensing data could be the time series analysis as applied for the detection of deforestation events (Kuemmerle et al., 2013; Joshi et al., 2016). Such information would enable improvements of the current implementation as well by e.g. deriving mowing events depending on grass biomass or growing degree days.

The role of heterogeneity in livestock distributions and grazing activities is currently not considered in the model. In the current implementation, the grass leaf biomass is reduced by a certain amount each day (paragraph 2.4.3). Ideally, the modeling

approach should capture the roaming of the livestock by reducing some plants completely, covering some with manure, trampling others and undisturbed growth of the remaining. An improvement of a globally applicable approach could distinguish the frequency of leaf biomass removal depending on the livestock density and prolong the interval from daily to weekly or monthly time periods.

Implementing more realistic grassland management options in DGVMs would help to improve analyses of future changes in the biogeochemistry of grasslands and their implications for sustainable land management. While we have focused on carbon, the importance of differences in grassland management options still needs to be assessed for water and eventually nitrogen dynamics. Human interference in the carbon balance of grasslands worldwide can have non-linear effects which might not be captured correctly when only considering it as a human appropriation of NPP (HANPP, Erb et al., 2009). The HANPP concept is based on the assumption that carbon sequestration of natural vegetation is reduced by human activities (Erb et al., 2016a). However, our results show that in grasslands under light to moderate grazing, the feedback on productivity may lead not only to higher carbon sequestration but also higher storage of carbon in the soil for a longer period. Especially with increasing livestock densities, we find a substantial feedback on NPP and soil carbon stocks. As such, these effects should be accounted for when assessing human impacts on the carbon dynamics on pasture areas. Applying an approach as we introduced and applied in this study would allow to disentangle the effects of climatic changes and changes in management on grassland which would help to reduce uncertainties and would allow for assessing options for adaptation to climate change.

Whereas our implementation of grassland management options is a necessary first step that establishes a major improvement in modeling capacities, the list of Kipling et al. (2016) suggests a range of further improvement strategies for grassland modeling and climate change impact assessments with these. It remains to be tested which of these suggestions are suitable for an implementation in a global-scale model and how sensitive modeling results are to these changes.

## 5 Conclusions

The presented implementation of grassland management in the DGVM LPJmL captures the substantial diversity of possible management practices on grasslands. Our results highlight the importance of management to understand and quantify feedbacks between biomass removal on global pastures and net primary production (NPP), carbon fluxes and soil carbon dynamics. We investigated the effect of different management practices on the global terrestrial carbon budget and found that yield and productivity of herbaceous plants affect changes in soil carbon under different climatic conditions that are consistent with regional studies and theory. Moreover, we can reproduce many non-linear and climate-dependent effects of livestock density and grazing intensity on biochemical cycles, as evidenced in various field studies. The magnitude of simulated impacts of the proposed grassland management options on biochemical processes and fluxes underlines the relevance of grassland management for assessing implications of agricultural activities for the global carbon balance.

Our results on the distribution of livestock density that triggers maximum grass harvest, as well as the maximum livestock density that can be supported by local grass production, quantify the influence of local climatic conditions on agricultural productivity. We also show that these practices that exploit the full biomass use potential have significant effects on soil carbon

stocks. A comparison of simulated grass harvests under different grazing options with European grassland productivity (Smit et al., 2008) reveal that best agreement with observed grass yields can be achieved assuming heterogeneous livestock densities.

Managed grasslands are still heavily under-researched in terms of global distributions of grazing livestock and wild herbivores and the implications of overgrazing in boreal and polar regions. With the model extension presented in this paper, the DGVM LPJmL can also contribute to the assessment of the ecological 'hoofprint' of livestock. Here, simulations of potential grass yields and the effects on soil carbon stocks may help to frame guidelines for sustainable grassland management and to better understand the implications of livestock production and climate mitigation targets.

## 6   Code availability

The code of the model LPJmL is available upon request for review purposes and for collaboration projects. Information on model versions and the download procedure can be found under https://www.pik-potsdam.de/research/projects/activities/biosphere-water-modelling/lpjml/versions.

## 7   Data availability

Simulation results and the code for the analysis are available upon request.

## Appendix A: Maps of harvesting option results

### A1  Default option $D$

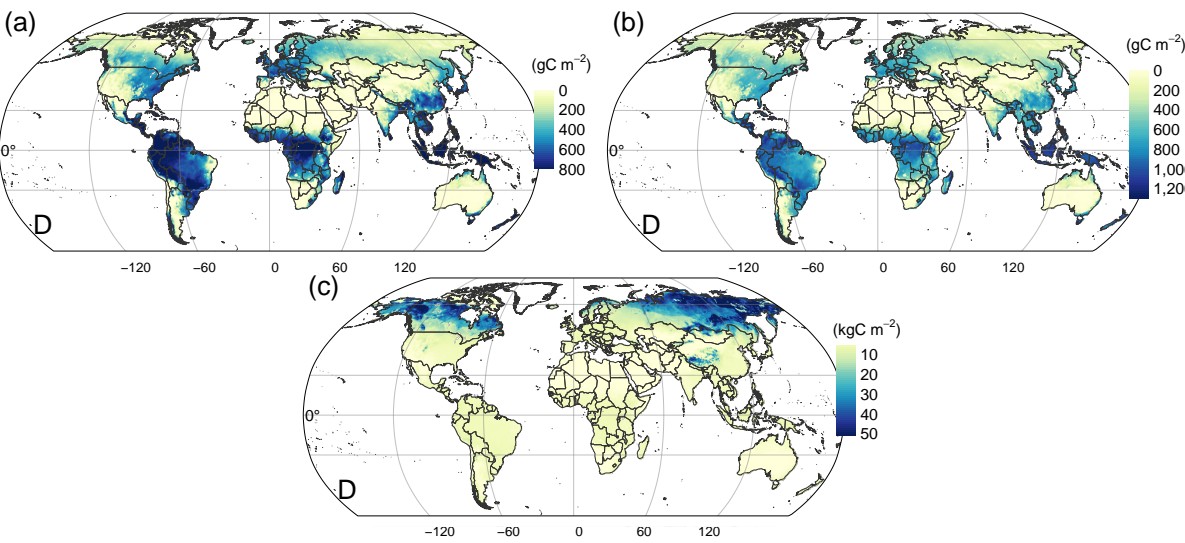

**Figure A1.** Simulation results for option $D$ averaged over 1998 to 2002, a) grass harvest (gC m$^{-2}$), b) NPP (gC m$^{-2}$) and c) soil carbon (kgC m$^{-2}$).

### A2  Mowing option $M$

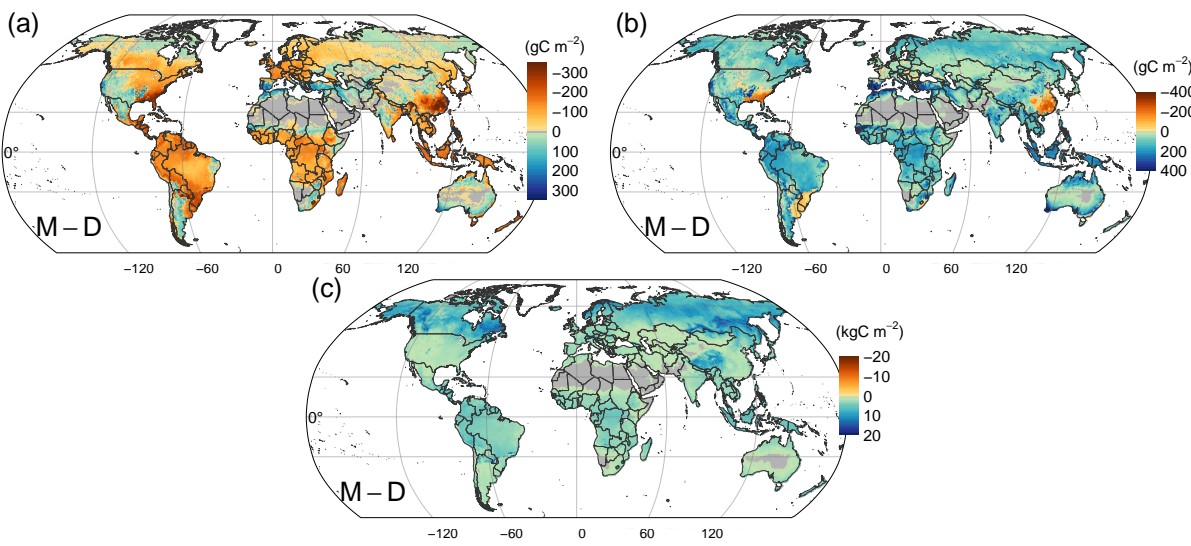

**Figure A2.** Simulation results for option $M$ averaged over 1998 to 2002, a) grass harvest difference to option $D$ (gC m$^{-2}$ a$^{-1}$), b) NPP difference to option $D$ (gC m$^{-2}$ a$^{-1}$) and c) soil carbon difference to option $D$ (kgC m$^{-2}$).

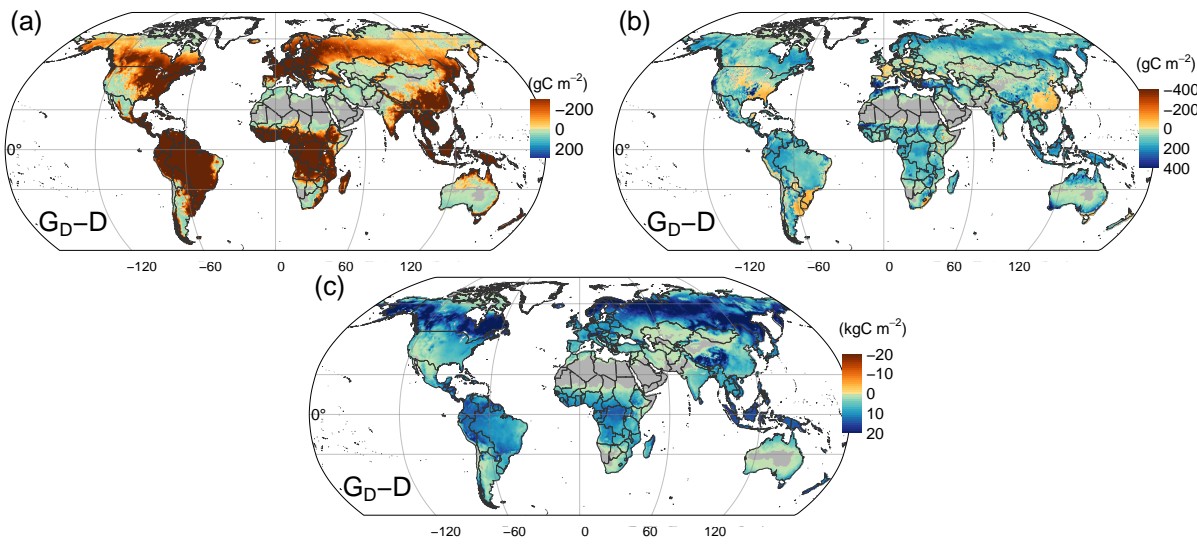

**Figure A3.** Simulation results for option $G_D$ averaged over 1998 to 2002, a) grass harvest difference to option $D$ (gC m$^{-2}$ a$^{-1}$), b) NPP difference to option $D$ (gC m$^{-2}$ a$^{-1}$) and c) soil carbon difference to option $D$ (kgC m$^{-2}$).

## A3 Daily grazing option $G_D$

## A4 Rotational grazing option $G_R$

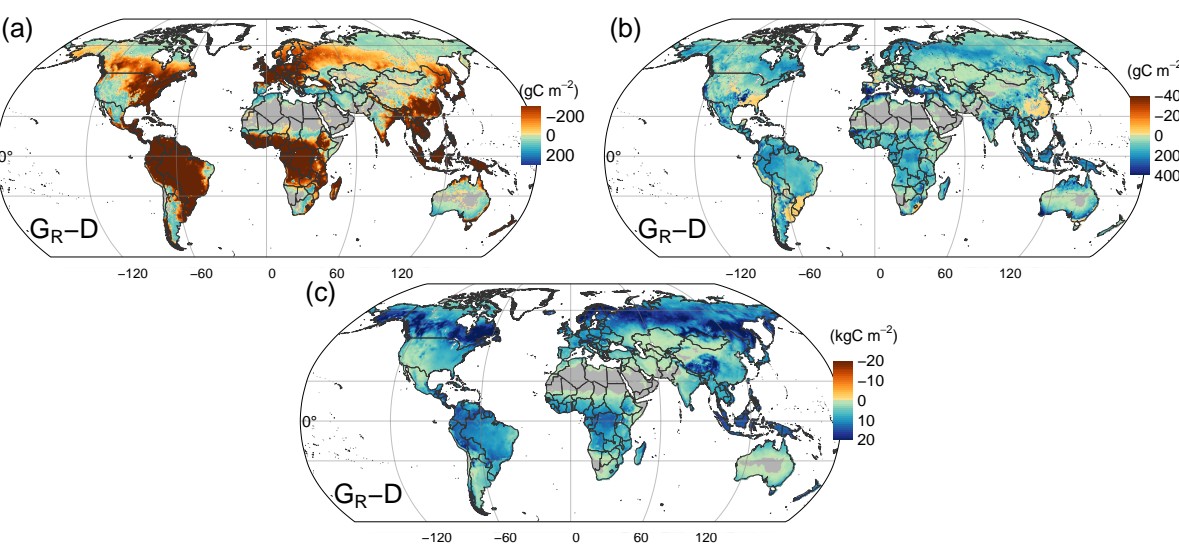

**Figure A4.** Simulation results for option $G_R$ averaged over 1998 to 2002, a) grass harvest difference to option $D$ (gC m$^{-2}$ a$^{-1}$), b) NPP difference to option $D$ (gC m$^{-2}$ a$^{-1}$) and c) soil carbon difference to option $D$ (kgC m$^{-2}$).

*Author contributions.* SR, JH, and CM designed the model improvement which was implemented by SR, JH and JtR. PAL, LB, EBP, IW and ABo contributed to parameter specification and design of the experiments. SR performed model simulations and analysis. SR wrote the manuscript with contributions from all co-authors.

*Competing interests.* We declare non competing interests.

5 *Acknowledgements.* SR and CM acknowledge financial support from the MACMIT project (01LN1317A) and the Kulunda project (01LL0905L) funded through the German Federal Ministry of Education and Research (BMBF) and support by Hermann Lotze-Campen and the Landuse Working Group at PIK. IW and BLB received funding from the European Union's Horizon 2020 research and innovation program under grant agreement No 652615 (SUSTAg FACCE-JPI). EBP, JtR and PAL acknowledge the financial support of the organisation "Statutory Research in Nature and Environment" in Wageningen. We enjoyed discussions and help from Bernhard Schauberger, Dieter Gerten, Yvonne 10 Jans, Femke Lutz, Sara Minoli and Stephen Wirth.

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
