# Peer review of "Modeling vegetation and carbon dynamics of managed grasslands at the global scale with LPJmL 3.6"

_Geoscientific Model Development, 2017_

## Referee Comment (RC2) · Anonymous Referee #2 · 6 Apr 2017

Rolinski et al., 2017

In this study, the authors have implemented four grassland management schemes in the model LPJml, and run global simulations with these four schemes varying some of the parameters. The authors aim to demonstrate the need for DGVMs to include grassland management because its impact on NPP and soil carbon. For this they analyze the effects of each grassland management system on three variables, grass harvest, NPP and soil carbon in a bioclimatic context depending on average temperature and precipitation. In a second time, the analysis focuses on potential applications of the modeling schemes and a comparison of the simulations with European data provides a validation of the order of magnitude of grass yields. Then the authors derive maximum livestock grazing densities.

**General comments The modeling approach and the scenarios defined are interesting and the results can bring some light into the role of grassland management on ecosystems functioning. However, the description of the modeling approach and underlying processes are not always very clear and should be more linked to the simulations' results. The description of the model is not very detailed and is disconnected from the results that never link the simulations' results with the model's structure causing these results. This link needs to be stronger. The authors argue for the lack of data for model validation but still there should be an effort in explaining how parameters were selected (no mentioning of calibration) in the absence of validation data.**

Also, the presentation and description of the results are sometimes too superficial and should be improved for the reader to get the full benefits from this study. In particular, the study is lacking a proper discussion section that explains and interprets the results that are so far only shown and described in a raw fashion.

Also the abstract is somewhat misleading on the main results of the paper. The main result highlighted in the abstract is not the main result developed in the results section. Also, the comment on application of LPJmL for global meat production seems too far a perspective to be in the abstract.

Overall, this work done for this study can be interesting for the geoscientific modeling community but efforts must be made in results presentation

**Specific comments**

-Methods The stand concept used to define vegetation types is not described. What characteristics are homogeneous within a stand that are not within a PFT? In equation (1), what are exactly the variables L, R and lr. If they are as described in the text above, then L/r=R so R-L/lr=0. A deeper explanation of this equation is needed. Also it is not clear how the optimal leaf to root ratio plays a role. In equations (3)-(6), parameter

values are set to 1 without stating it in the equations making it confusing. Please write in the equation when you put alpha_leaf=1 and Nind=1, and explain in the text what is the meaning of setting these values to 1. Also, for lrp, on line 14p6, it is said that it is PFT-specific and set to 0.75. Is it 0.75 for all grassland ecosystem?

-Calibration Many parameters are used in the model and there is no reference as to where they come from. An example is in equations (3)-(6).

-Results About the presentation of results, the graphical items used to display the results sometimes make the reading hard to follow. The maps and temperature/precipitation (T,P) graphs are redundant. Maps are not bringing any additional information since the modeling is too simplified to give realistic values (no map of grassland, spatial homogeneity of management practices, no fertilization & irrigation, no water feedback) except for some reader who are looking for specific values in some data. Maps can even be misleading. For example on fig.5b, the areas for which the difference is negative seem unrelated when in the T,P plots of fig. 6b, it is clear that there is a similar process in these regions due to their bioclimatic conditions. Moreover maps are difficult to compare visually to each other. They should be moved to appendix. To allow for mental representation of geographical distribution, the separation lines between T,P areas appearing in fig. 2 should be reported in later T,P plots to allow for rapid bioclimatic regions differentiation. Also, even with maps in appendix, because maps and T,P plots show the same information (even if aggregated by deciles of precipitation and temperature in the latter) it would help the reader to use the exact same color scale for both families of plots showing a similar variable. The sequential green/blue color scale used for T,P plots in fig 4 is less likely to introduce an artificial visual bias than the divergent color scale in fig 5.

In several occasions in the text, grass yield and soil carbon patterns are explained from their relation to NPP. The described relationship is not visible from the data shown making the text impossible to follow for the reader. Graphs of yield versus NPP and soil carbon versus NPP would help convince the reader of the significance of the trends

and relationships described.

Color scale used in the difference plots is counter-intuitive with increase in cold colors and decrease in warm colors. Also, the color scale is too close from the one with absolute values to see right away that what is plotted is a difference. Fig. 5a and 10a are not described, what is described and should be shown (in appendix) is the difference in harvest between scenarios for consistency with text.

The authors attempt to compare their simulation to regional data in Europe. This exercise is very ambitious given the level of simplification of the model, in the spatial homogeneity, diversity of scenarios and processes involved. However it can be an informative comparison if well explained and described. For example, the reason for choosing to compare only the highest harvest GD simulations with data is not explained. If it is supposed to be the more realistic given current practices it should be justified. An interesting result would be to show which management setting leads to the best simulation in each subregion and try to explain it.

About the results in general, the article lacks an analysis of the results. The discussion section is about effects on soil carbon, uncertainties and further developments in the modelling approach. If all these discussion points are interesting, after a very descriptive results section, the reader is also expecting an interpretation of the results, as a full discussion with explanation of the underlying processes and implications. For example, what processes in the model drive the feedbacks? Some of the feedbacks are the simple expression of the relationships coded in the model and this should be identified and its realism described. What is the reason for the pattern in fig. 6,8 & 11 (climatic area 10<T<20 & 1200<P, pattern mentioned but not explained in the text)? Does the soil parameterization (texture) play a role in the results or other ignored variables?

**Editorial comments In introduction the abbreviation Mio for Million is not the conventional one. Fig. 1 : axis labels need to be more explicit. P4l23 typo : "1 and 1" P6 text in 2.3.1 introduces 2.3 but not 2.3.2 P6L3 sentence not clear P6l28 'are used' instead**

[Figure]

of 'is used' P13L4 the sentence Âńaverage grass yield and soil carbon under these conditions are not substantially different Âż is confusing. It sounds like grass yield and soil C are the same. Rephrase, maybe use 'homogeneous'.

Fig. 9 please make the figure visually lighter by using only one color bar per row.

P17L21 rewrite sentence

p19L11 not clear, rewrite sentence

p19L16 'low correlations', 'high standard deviations and RMSD'. Give the numbers.

P22L05 why compare LSUmax to scenario M and not default D as in all the rest of the manuscript ?

---

## Author Comment (AC1) · 21 Jun 2017

**Overview**

*This manuscript describes the implementation of 3 new grassland management options into the dynamic vegetation model LPJmL. The new grassland management options were set up in order to model the major ways of how grasslands are managed worldwide. These options were parametrized using reference values from the literature. Then, global simulations of LPJmL over the period 1901-2009 with a daily time step are conducted. The results show global maps of simulated grass growth (NPP), harvest and soil carbon values for the different grassland management options. Then,*

*observed grassland harvest data for Europe were used for comparison to simulated values. Lastly, a sustainable potential livestock density map is drawn, depicting the optimal livestock unit density which allows maximum simulated harvest. In the last section (discussion), global results are discussed and compared to other findings in the literature, and perspectives for global simulations are presented.*

*The paper is nicely written and well structured. It is well-sourced with relevant references in the introduction, the methods (where references values are used for designing the management options) and in the discussions. Figures are well done, although a better choice of some colour scales could be made. It presents new interesting features for grassland modelling with LPJmL, and for the wider DVGM community. Beside the particular issue of the calibration/validation, this paper is nearly ready for publication in GMD.*

We thank the reviewer for the constructive feedback. Our responses are inserted below, following their original comments.

**General comments**

*No calibration/validation: This paper presents an implementation of grassland management techniques but no calibration of the model parameters is presented. Though a database of direct observations of grassland productivity at the global scale is not existing, some indirect global products may help to somehow calibrate and validate the approach. For instance, LSUmax densities appears too optimistic in arid areas compared to existing database on livestock densities (see next remark). In particular, a global map of grassland areas (from Globcover for instance) might be used to validate the extent of grassland in arid areas. Validation is presented but only for Europe and only for one of the grazing options. This should be addressed or better discussed in the manuscript.*

We agree that an evaluation of model results is important and not well covered in the paper due to lack of good reference data. We will discuss this more thoroughly in the

discussion section (section 4.2). For similar reasons, we chose not to calibrate any of the literature-based parameters. The observed extent of grassland is not a good reference as it is a) subject to uncertainties in delineating it from forests/shrubland/fallow land and b) the model does not predict the extent of grassland but its productivity and carbon/water balance. In the simulations conducted here, we prescribe 100 % grassland to all land area to analyze spatial patterns of grassland productivity and dynamics rather than identifying their actual extent.

Nevertheless, we see the need to make parameter choices better comprehensible and compare results better to existing estimates. We will tackle this by 3 major changes:

- We will include better references to the parameter discussion in 4.2 at the beginning of the results section such as:
  'The effect of the harvest options are described for grass yield, NPP, and total soil carbon of the 3 m soil column and analyzed with respect to the underlying processes (see also discussion on strengths and weaknesses of the chosen approach in section 4.2).'

- Section 3.2 will be rewritten completely so that the comparison with reported grass yield now serves the purpose to evaluate the spatial distribution of different management options and the probability of the applied management for the reported values. We test 3 hypotheses, namely if 'European grass harvest

  1. can be achieved by grazing animals only,
  2. is determined by management and only to a minor degree by climate and
  3. per geographical entity, a dominant management option can be identified that results in similar harvest values as reported.'

by analyzing different selections from the simulation results. This allows to check the value range within the results and the plausibility of the distribution of management options in Europe although a detailed dataset does not exist.

- The calculation of potentials will be complemented as described below.

*Figure 14 & Sustainable potentials: I'm rather surprised of the $LSU_{max}$ densities values in the global map of Fig. 14. It seems that $LSU_{max}$ densities are overestimated in many arid regions. For instance, all of Morocco area has a $LSU_{max}$ value around 1, while this country is partly covered by arid deserts where no cattle grazing is possible (except in small irrigated areas). IMHO, $LSU_{max}$ values seems also overestimated in Lybia, central Australia & Saudi Arabia. I understand that the LPJ simulations do not take into account all processes involved in land degradation (such as historical overgrazing) but the grasslands production in arid regions seems clearly overestimated. In order to quantify this, it would be interesting to compare grassland production and/or livestock densities with other database. Robinson et al. PloS One, Mapping the Global Distribution of Livestock, 2014 and the companion website http://livestock.geo-wiki.org/ provides livestock densities data worldwide. I understand that Fig. 14 presents a potential maximal LSU (i.e., with 100% of land use affected to grasslands), but a comparison of your findings to the geo-wiki database (or others) would allow to somehow validate your findings about sustainable potentials. Maybe your model parameters should be adapted in order to reflect a better view of grass production in arid areas.*

We see that the term $LSU_{max}$ was ambiguously used and not well framed. These numbers represent the livestock density under which the maximum grass yield is obtained and in most areas this number is not the livestock density that can be fed by the grass, i.e. that livestock density would need supplement feed from elsewhere. This is especially true in the arid regions mentioned. Here, short rain events allow grass growth for a short vegetation period. Grass harvest under these conditions are highest with a moderate to high LSU, owing to the feedback from grazing on productivity (LAI, respiration). But the harvest itself is very low (mean $\pm$ standard deviation are for Libya $9\pm15$, for Morocco $73\pm42$, for Australia $80\pm73$ and for Saudi-Arabia $6\pm9$ gC m$^{-2}$). Therefore, we choose a different name for this variable with $LSU_{harv}$ suggesting that this livestock density is optimized with respect to the obtained harvest.

We will adjust the description in section 2.6.2 with the new title
'Determination of harvest potentials under $G_D$'
and describe the number of LSU that can actually be fed exclusively LSU$_{feed}$ at the end of the paragraph with
'To obtain the maximum livestock density that can be fed with the grass available throughout the year, the maximum livestock density is chosen under which harvest meets the demand. LSU$_{feed}$ is thus maximized with respect to the livestock that can be supported by the local grass production.'

For LSU$_{feed}$, the result is included as Figure 14b (Fig. 1) and described in the text with 'The distribution of the maximum livestock density that can be fully supported by the local grass production (LSU$_{feed}$) (Fig. 14b) differs from LSU$_{harv}$ especially in arid regions like inner Australia, North Africa, western North America and the Middle East where values close to 0 LSU ha$^{-1}$ are derived for LSU$_{feed}$. Also in polar areas in North America and Asia (region 1 in Fig. 2), values for LSU$_{feed}$ are reduced considerably. Even though grass productivity can be high in parts of the year, this is not sufficient to continuously supply the feed demand of higher livestock densities.'

In addition, we will compare LSU$_{max}$ and LSU$_{feed}$ to actual livestock densities LSU$_{fao}$ as given by Robinson et al. (2014) as Figure 14 c (Fig. 2). These combine all ruminant animals and livestock production systems so that the distribution of pastoral and mixed systems has to be considered for the analysis. We will also discuss possible mismatches where local feed supply is not sufficient, as also highlighted by Herrero and Thornton (2013).

**Specific comments**

- *P3L1: Title of section 1.2 (Representation of managed grasslands in DVGMs) is misleading because the section only states about representation of managed grasslands in LPJ and ORCHIDEE (that partly originates from LPJ). There are no discussions about how this is done in other DVGM, if any.*

To our knowledge, only ORCHIDEE and LPJmL do have managed grassland representations. We state that now clearly at page 3 line 14:

'To the knowlegde of the authors, managed grasslands are not represented in further DGVMs.'

If required or appropriate, we include the chosen approaches of DGVMs such as CLM or JSBACH for pasture areas.

'The Community Land Model (CLM) treats pasture in the version with representation of agricultural activity (CLM-Crop, Drewniak et al., 2015) as natural grassland without harvest procedure whereas JSBACH (Reick et al., 2013) simulates fire disturbances on grassland but no other harvest is taken into account.'

- *P3L8: There is no adequate description of the way managed grassland was simulated so far in LPJ. The statement "It has been represented as grassland ecosystem with human management" is too vague.*

We agree although we have a little problem here. In the description of the implementation of agricultural activities by Bondeau et al. (2007), the focus was on the description and parametrization of the crop functional types. Grassland was introduced as agricultural activity with a productivity-dependent removal from the aboveground carbon pool and daily allocation. There were no specific descriptions of e.g. allocation rules. When the code was transferred from C++ to C, the implementation of managed grassland was changed but not documented in a further publication. Thus, we aim at a first detailed description of managed grassland in LPJmL in this paper without going into details of the prior unpublished implementation. We would like to avoid a detailed description of the now obsolete implementation. Obviously, this approach is not satisfying so that we will exchange lines 8 to 10 in section 1.2 to

'It has been implemented as grassland ecosystem with a harvesting rule. Grass plants grew and competed for light and water. Harvest was depending on grass productivity solely. When more than 100 gC m$^{-2}$ was assimilated since the last

harvest event, half of the aboveground carbon pool was removed. Assimilated carbon was allocated to leaves and roots prior to harvest and at the end of the year following the rules for natural grasses.'

- *P3L15-19: At this stage of reading, there is a contradiction about the number of management options that were implemented: is it 3 or 4? The reader understands only later that there 3 new options + 1 default option.*
  We will clarify from the beginning that we are describing 4 management options. The default option that was applied before is also reformulated within the development of the 3 new options so that they should be mentioned as a group of 4 options. The only distinction of the default setting is the possibility to apply it without further knowledge on the distribution of grassland management. This will be taken up in the entire manuscript.

- *P3: The objective(s) of the manuscript is (are) explained on L27-32 but they could be more clearly defined, maybe using bullets points. Is it to test/calibrate/validate the implementation of new functionalities in the simulation of managed grasslands? To evaluate the importance of accounting for grassland management in NPP global estimation?*
  We see the point and will exchange lines 27-30 by
  'Without being able to represent actual grassland management at this stage, we are aiming with this implementation at the following objectives:

  – a much better representation of the diversity in grassland management at the global scale in model simulations of agricultural productivity and biogeochemical cycles,

  – a demonstration of the role of grassland management for biogeochemical simulations by analyzing the effects on Net Primary Productivity (NPP) and soil carbon stocks,

- – an assessment of potentials of agricultural productivity by determining maximum harvest and the associated livestock densities with and without the condition of maintaining soil carbon stocks.

- – to evaluate model performance by comparing simulated harvest with a European data set (Smit et al., 2008) and potential livestock densities with data from the Gridded Livestock of the World v2.0 (Robinson et al., 2014).'

- *P6L24-30: The way the model reacts after an harvest event is central to the modelling of mowing. More details or explanations on the feedbacks could be interesting. For instance, how much time does it takes to the photosynthetic activity to recover from the cut (in general)? Is some transfer of C from roots to leaves after a cut simulated?*

The recovery period depends on the climatic conditions since they determine net primary productivity from the actual leaf carbon content. We follow the common rules for allocation of assimilated carbon as described in section 2.3.2 which have two consequences for the period after a mowing event: very low leaf carbon reduces NPP but lower leaf biomass that still intercepts large fractions of incoming light may also enhance NPP, as the maintenance respiration is reduced more strongly than the light interception. We will extend the methods section 2.3.2 by a more detailed description of the water limitation of photosynthesis and include the sentences:

'After a harvest event, leaf carbon and thereby $lr$ is reduced. Carbon allocation in the following period will try to reestablish the actual leaf to root mass ratio $lr$. Depending on the water supply to demand ratio, the assimilated carbon is incorporated more or less to the leaves so that the actual water conditions determine the recovery time of the leaves. A 10 % reduction of the water supply alone would result in a slower recovery time of several days and leaves would have less carbon when the new $lr$ is established. Even more important is the feedback on primary productivity connected to the leaf carbon content.'

and at the end of the paragraph:

'This dependency of light absorption and photosynthesis on leaf carbon content leads to a negative feedback of harvest on absorbed radiation. When leaf carbon is reduced to 50 %, the reduction of fAPAR is about 30 % for $L = 100$ gC m$^{-2}$ and is diminished to 2 % for $L = 500$ gC m$^{-2}$.'

- *P6L27 or eq. (3): SLA units are missing.*
  We will include 'SLA in m$^2$ gC' in the brackets in line 27.

- *P7L8-10: IMHO, many pastures worldwide cannot be mowed (by machines) because of impractability (very steep pastures, presence of stones/trees, non-portable soil because it is too wet, : : :).*
  Thank you for the suggestion. We change the second sentence in line 9 to:
  'When mowing is not an option due to the steepness of the landscape, soil wetness or obstructing trees and boulders, grazing by smaller ruminants might still be possible but mowing and grazing by livestock are often used in combinations.'

- *P10 & Fig. 2: What are the rationales behind this climate classification? Why not using classical classification such as the one of Köppen?*
  With this classification, we try to connect the maps (e.g. Fig. 3) and the figures (e.g. Fig. 4). The chosen thresholds are motivated by the values in the climate response figures. Especially region 6 evolved because there the NPP increase with increasing livestock densities (Fig. 9) prevailed. I am not aware that the Köppen Geiger classification would serve the same purpose being much more detailed and including seasonality characteristics. To better motivate the chosen thresholds we change the sentence page 10 lines 22 to 24 to
  'Ranges of temperature and precipitation for which grassland management results in similar changes in the carbon dynamics are classified as bioclimatic regions (Fig. 2 a) only for a better visualization of locations of similar climatic conditions (Fig. 2 b).'

- *P23 L29-30: I suggest to move this sentence at the beginning of the section 4.1.*
  Done.

- *P23 L33: I would not say that the comparison with European grassland data showed "good agreement".*
  We see the point and have 2 remarks:

  – Unfortunately, the determination of the average harvest per geographical unit was incorrect and mostly a bit too low. With the corrected version, the correlation in Figure 12c increases from 0.8 to 0.88 and the standard deviation from 65 to 78 gC m$^{-2}$ which is closer to the presented data. Additionally we include a third selection of simulation results that gives a correlation of 0.9 with a standard deviation of 93 gC m$^{-2}$.

  – We rewrite the whole section and concentrate on the plausibility of management distributions (see above) because the assumptions on management were homogeneous in the simulations which is definitely not the case in Europe. We state that in the beginning with
  'Since management assumptions for the simulations were spatially homogeneous and management in Europe is known to vary spatially as well as temporally, we use the comparison to find out whether climate- and management-induced variations in grass harvest can be captured by the applied options.'

- *P25 – 4.3 Further developments: This paper lacks of further validation. I would suggest to add in this section as a perspective a short review of literature about the use of remote sensing data to further validate the implementation of grassland management options. For instance, although I did not find adequate references, change detection techniques based on remote sensing data can detect hay mowing. More well-known is the use of vegetation indices derived from remote sensing data to estimate standing biomass.*

We thank the reviewer for mentioning this point. We know of the approach of the group of Patrick Hostert at Humboldt University Berlin to identify the timing and extent of deforestation events from satellite data (Kuemmerle et al., 2013; Joshi et al., 2016) and were in contact to explore possibilities to extend the methods on grasslands. So far, we are not aware of already undertaken attempts to relate remote sensing analysis to grass harvest but we would be very interested in such approaches. Since we also did not find respective references, we would like to include this aspect in the first paragraph of discussion section 4.3:
'A promising approach to extract cutting events on grassland from remote sensing data could be the time series analysis as applied for the detection of deforestation events (Kuemmerle et al., 2013; Joshi et al., 2016).'

- *P 26 - 5. Conclusions: The "Conclusions" section is short and does not present key numerical results. This could be improved.*
  We will extend the conclusions and relate the discussed topics back to the now formulated objectives of the manuscript. The outline would be like this:
  'The presented implementation of grassland management in the DGVM LPJmL captures the substantial diversity of possible management practices. Our results highlight the importance of management to understand and quantify feedbacks between biomass removal on global pastures and net primary production (NPP), carbon fluxes and soil dynamics. We investigated the effect of different management practices on the global terrestrial carbon budget and found that yield and productivity of herbaceous plants show feedbacks with the development of soil carbon under different climatic conditions that are consistent with regional studies and theory. Moreover, we can reproduce many non-linear and climate-dependent effects of livestock density and grazing intensity on biochemical cycles, as evidenced in various field studies. The magnitude of simulated impacts of the proposed grassland management options on biochemical processes and fluxes underlines the relevance of grassland management for assessing implication of agricultural activities for the global carbon balance.

Our results on the distribution of livestock density that triggers maximum grass harvest, as well as the maximum livestock density that can be supported by local grass production, quantify the influence of local climatic conditions on agricultural productivity, where we additionally consider the impact of these practices that exploit the full biomass harvest potential on soil carbon stocks. Comparison of simulated grass harvest under different grazing options with European grassland productivity (Smit et al., 2008) reveal that best agreement with observed grass yields can be achieved assuming heterogeneous livestock densities.

Managed grasslands are still heavily under-researched in terms of global distributions of grazing livestock and wild herbivores and the implications of overgrazing in boreal and polar regions. With the model extension presented, the DGVM LPJmL can also contribute to the assessment of the ecological 'hoofprint' of livestock. Here, simulations of potential grass yields and the effects on soil carbon stocks may help to frame guidelines for sustainable grassland management and to better understand the implications of livestock production and climate mitigation targets.'

- *Fig. 3: Suggestion: The colour scale of figure 3 is a divergent colour scale but should be changed into a sequential colour scale, as in Fig. 12. Sequential colour scale might be easier to interpret, are color-blind & black/white print friendly, and fits to the grass harvest, NPP and soil carbon variables, which are sequential variables. However, the divergent colour scales in Figs. 5 b-c, 7 b-c, 10 b-c are OK since differences in NPP and soil carbon between options are divergent variables.*

We will change the color scale for all maps showing absolute values (now figures 3, 5a, 7a, 10a, 14).

- *Fig 4: Incoherence of scales: Fig. 4 a) scale for grass harvest is from 0 to 500 gCm-2 while Fig. 3 a) scale is 0-800 gCm-2.*

This incoherence is caused by the averaging of values shown in Fig. 4a. The scales are chosen to be representative for comparable variables (e.g. Figs. 5b, 7b and 10b or 5c, 7c and 10c) and deviating values are given in the text (e.g. page 12 line 24). To compute values shown in the climate response plots, these are averaged for same temperature and precipitation bins.

**Editorial comments**

- *P3L21: ": : : to that in (Bondeau et al., 2007), ..." should be ": : : to that in Bondeau et al. (2007), ..."*
  Right, done.

- *P3L29: Suggestion (not sure): "...we compare the data with..." should be ": : : we compare the simulations with..."?*
  Right, done using the term 'simulation results'.

- *P17L7: Seems there is a space missing between "manure." and "When".*
  Right, done.

**References**

Bondeau, A., Smith, P. C., Zaehle, S., Schaphoff, S., Lucht, W., Cramer, W., and Gerten, D.: Modelling the role of agriculture for the 20th century global terrestrial carbon balance, Global Change Biology, 13, 679 – 706, doi:10.1111/j.1365-2486.2006.01305.x, 2007.

Drewniak, B. A., Mishra, U., Song, J., Prell, J., and Kotamarthi, V. R.: Modeling the impact of agricultural land use and management on US carbon budgets, Biogeosciences, 12, 2119 – 2129, doi:10.5194/bg-12-2119-2015, 2015.

Herrero, M. and Thornton, P. K.: Livestock and global change: Emerging issues for sustainable food systems, Proceedings of the National Academy of Sciences, 110, 20 878 – 20 881, doi:10.1073/pnas.1321844111, 2013.

Joshi, N., Baumann, M., Ehammer, A., Fensholt, R., Grogan, K., Hostert, P., Jepsen, M. R., Kuemmerle, T., Meyfroidt, P., and Mitchard, E. T.: A Review of the Application of Optical and Radar Remote Sensing Data Fusion to Land Use Mapping and Monitoring, Remote Sensing, 8, 70, doi:10.3390/rs8010070, 2016.

Kuemmerle, T., Erb, K., Meyfroidt, P., Müller, D., Verburg, P. H., Estel, S., Haberl, H., Hostert, P., Jepsen, M. R., Kastner, T., Levers, C., Lindner, M., Plutzar, C., Verkerk, P. J., van der Zanden, E. H., and Reenberg, A.: Challenges and opportunities in mapping land use intensity globally, Current Opinion in Environmental Sustainability, 5, 484 – 493, doi:10.1016/j.cosust.2013.06.002, 2013.

Reick, C. H., Raddatz, T., Brovkin, V., and Gayler, V.: Representation of natural and anthropogenic land cover change in MPI-ESM, Journal of Advances in Modeling Earth Systems, 5, 459 – 482, doi:10.1002/jame.20022, 2013.

Robinson, T. P., Wint, G. R. W., Conchedda, G., Van Boeckel, T. P., Ercoli, V., Palamara, E., Cinardi, G., D'Aietti, L., Hay, S. I., and Gilbert, M.: Mapping the global distribution of livestock, Plos One, 9, e96 084, doi:10.1371/journal.pone.0096084, 2014.

Smit, H. J., Metzger, M. J., and Ewert, F.: Spatial distribution of grassland productivity and land use in Europe, Agricultural Systems, 98, 208 – 219, doi:10.1016/j.agsy.2008.07.004, 2008.

———————————————————

[Figure]

**LSU at pure grass feed**

(b)

LSU ha$^{-1}$

0.0
0.5
1.0
1.5
2.0

**Fig. 1.** Distribution of maximum livestock densities that can be fed purely on the local grass harvest (LSU$_{feed}$ in LSU$\sim$ha$^{-1}$) under harvest option G$_D$ averaged over the years 1998 to 2002.

[Figure]

FAO LSU

(c)

0°

−120    −60    0    60    120

LSU ha$^{-1}$
0.0
0.5
1.0
1.5
2.0

**Fig. 2.** Distribution of ruminant livestock densities as reported by FAO (LSU$_{fao}$ in LSU~ha$^{-1}$).

---

## Author Comment (AC2) · 21 Jun 2017

**Overview**

In this study, the authors have implemented four grassland management schemes in the model LPJml, and run global simulations with these four schemes varying some of the parameters. The authors aim to demonstrate the need for DGVMs to include grassland management because its impact on NPP and soil carbon. For this they analyze the effects of each grassland management system on three variables, grass harvest, NPP and soil carbon in a bioclimatic context depending on average temperature and precipitation. In a second time, the analysis focuses on potential applications of the

modeling schemes and a comparison of the simulations with European data provides a validation of the order of magnitude of grass yields. Then the authors derive maximum livestock grazing densities.

Thank you for the concise overview. The last item of deriving potentials was misleading also for another reviewer so that we changed and extended the approach (description under specific comments). Our responses are inserted below, following their original comments.

**General comments**

The modeling approach and the scenarios defined are interesting and the results can bring some light into the role of grassland management on ecosystems functioning. However, the description of the modeling approach and underlying processes are not always very clear and should be more linked to the simulations' results. The description of the model is not very detailed and is disconnected from the results that never link the simulations' results with the model's structure causing these results. This link needs to be stronger.

We acknowledge that a storyline from methods over results to the discussion is only weak and should be much clearer linking underlying model assumptions to resulting trade-offs between productivity, harvest and carbon sequestration in the soil. Therefore, we include

1. more precise description of the methods including formulae. Therefore, P6L13 to P6L15 is exchanged with

'Partitioning of assimilated carbon  $B_I$  to leaves and roots is calculated in a way that a given leaf to root mass ratio is approximated. The PFT-specific parameter  $lr_p$  is 0.75 for both grasses (Sitch et al., 2003), i.e. that 0.75 times leaf carbon equals root carbon under optimal conditions.  $lr_p$  is scaled to the actual ratio lr(Eq. 1) with a measure of water stress (actual ratio of plant water supply  $W_{supply}$ to atmospheric water demand  $W_{demand}$ ) (Eq. 2 in Sitch et al., 2003). Under dry GMDD
conditions, this scaling results in a lower lr so that the allocation of assimilated carbon is shifted towards the roots.

$$lr = \max(0.25, lr_p \cdot W_{supply} / W_{demand})' \tag{1}$$

and more detail is given instead of P6L24 to P6L25:

'After a harvest event, leaf carbon and thereby lr is reduced. Carbon allocation in the following period will try to reestablish the actual leaf to root mass ratio lr. Depending on the water supply to demand ratio, the assimilated carbon is incorporated more or less to the leaves so that the actual water conditions and NPP determine the recovery time of the leaves. Without a feedback to primary productivity, a 10 % reduction of the water supply alone would result in a slower recovery time of several days and leaves would have less carbon when the new lr is established. Hence, even more important is the feedback on primary productivity connected to the leaf carbon content.'

Also the source of the parameters is given explicitly after P7L1:

'Some of the parameters have standard values such as  $\alpha_{leaf} = 1$  for grasses, i.e. grass leaves are assumed to be photosynthetically active for 1 year (Eq. (6) in Sitch et al., 2003),  $N_{ind} = 1$ , i.e. that one average individual is considered (as set in Sitch et al., 2003), and  $k_b = 0.5$ , as established in the literature and recently confirmed by Saitoh et al. (2012) (Eq. (7) in Sitch et al., 2003).'

The section now ends with an estimate for the effects of carbon removal from the leaves:

'This dependency of light absorption and photosynthesis on leaf carbon content leads to a negative feedback of harvest on absorbed radiation. When leaf carbon is reduced to 50 %, the reduction of fAPAR is about 30 % for L = 100 gC m-2 and is diminished to 2 % for L = 500 gC m-2.'

2. We rewrite most of the results section and end each subsection with an assessment of the processes leading to the simulated results. For the default option, GMDD
this would read like:

'The frequent mowing option D generates nearly optimum grass yield in all regions with a minimum productivity. As soon as regrowth occurs, leaf biomass is removed without an additional or residual flux into the soil. Productivity under high temperatures and precipitation is even enhanced by the harvesting because of the comparatively high residuals (Fig. 1) for high leaf biomass. The feedback of leaf carbon content to photosynthesis favours plant regrowth in these regions because leaves after the harvest are growing exponentially. Cold and low productive regions with low respiration and turnover provide the only environment in which high values of soil carbon are reached under option D.'

3. We include a better reference to the discussion of strengths and weaknesses of process formulations in the beginning of the results section:

'The effect of the harvest options are described for grass yield, NPP, and total soil carbon of the 3 m soil column and analyzed with respect to the underlying processes (see also discussion on strengths and weaknesses of the chosen approach in section 4.2). '

The authors argue for the lack of data for model validation but still there should be an effort in explaining how parameters were selected (no mentioning of calibration) in the absence of validation data.

We agree that although the lack of global benchmark data is a drawback, the parameter choices should be better motivated and the discussion on our parameters in relation to other approaches should be better visible. We will extend the text accordingly, see below for specific comments and section 4.2.

Also, the presentation and description of the results are sometimes too superficial and should be improved for the reader to get the full benefits from this study. In particular, the study is lacking a proper discussion section that explains and interprets the results that are so far only shown and described in a raw fashion.
We see that the combination of result description with a reference to the processes causing these results will enhance the readability and value of this section. We decided to improve the presentation of the results by rewriting the section and including a paragraph in each subsection about the underlying processes that lead to the simulated results (see item 2 above).

Also the abstract is somewhat misleading on the main results of the paper. The main result highlighted in the abstract is not the main result developed in the results section. Also, the comment on application of LPJmL for global meat production seems too far a perspective to be in the abstract.

We see that the abstract is not adjusted to the main results and the reference to global meat production a bit far-fetched. We will adjust the abstract accordingly.

Overall, this work done for this study can be interesting for the geoscientific modeling community but efforts must be made in results presentation.

Thank you for this overall assessment. In the following, we will try to meet your recommendations.

**Specific comments**

**Methods**

- The stand concept used to define vegetation types is not described. What characteristics are homogeneous within a stand that are not within a PFT? The concept is defined in section 2.2 (P4L29) and explained in more detail in section 2.2.2 (P5L27 to P5L31) and 2.3.1. We include references to the first description.
- In equation (1), what are exactly the variables L, R and Ir. If they are as described in the text above, then L/r = R so R L/lr = 0. A deeper explanation of this equation is needed. Also it is not clear how the optimal leaf to root ratio plays a
role.

We see that the description was not informative and adjust the definition of lr also with an equation (see item 1 on page 2). The parameter lr is the actual and changing target for carbon allocation between leaves and roots so that usually  $L/R \neq lr$  either because the water supply to demand ratio changed or the plant compartments changed after a harvest event. We include more explanation of the effect of the variability of lr when describing the effect of a harvest event and illustrate the feedback of leaf carbon reduction to photosynthesis at the end of the paragraph.

• In equations (3)-(6), parameter values are set to 1 without stating it in the equations making it confusing. Please write in the equation when you put  $\alpha_{leaf} = 1$  and Nind = 1, and explain in the text what is the meaning of setting these values to 1. Also, for lrp, on line 14p6, it is said that it is PFT-specific and set to 0.75. Is it 0.75 for all grassland ecosystem?

We remove the setting from the text, include the values and their consequences into the equations and add one sentence before the equations (see item 1 on page 2) to give the source of the chosen values. We hope that this was intended by the remark.

**Calibration**

• Many parameters are used in the model and there is no reference as to where they come from. An example is in equations (3)-(6).

We acknowledge that the background of the parameter choices has to be made clearer. Most of the parameter values are already described in the literature on the implementation of processes in LPJmL which is stated now clearly in section 2.3.1.The mentioned equations (now 4 to 7) are reported as published in Sitch et al. (2003) and we clarify now which equations and parameters are taken from
this publication.

For those parameters which are introduced for accounting for grassland management, values are estimated from literature. These are discussed in section 4.2 which is referred to in P7L17. We did not apply a calibration technique such as Bayesian parameter estimation or another fitting procedure because we did not find a reliable global dataset to be used in such an exercise. We admit that the chosen parameters may not be ideal but with this model development presented here, we aimed primarily at the representation of grassland management which in future can be made use of to derive spatial distributions of management options and intensities. One example of the potential combination of modeling and data analysis was presented by Chang et al. (2016) but goes beyond the scope of this paper. We hope to clarify this point when stating the aims and objectives of the implementation in exchange with P3L27 to P3L32 and explicitly point out that parameters are not calibrated here, which could be included in future work with appropriate reference data (e.g. in regional studies):

Without being able to represent actual grassland management at this stage, we are aiming with this implementation at the following objectives:

- comprehensive representation of the diversity in grassland management and in related feedbacks between biomass removal and primary productivity,
- demonstration of the role of grassland management for biogeochemical simulations by analyzing the effects on grass yield, Net Primary Productivity (NPP) and soil carbon stocks,
- assessment of potentials of agricultural productivity by determining maximum harvest and the associated livestock densities with and without the condition of maintaining soil carbon stocks.
- evaluation of model performance by comparing simulated harvest with an European data set (Smit et al., 2008) and potential livestock densities with
**Results**

• About the presentation of results, the graphical items used to display the results sometimes make the reading hard to follow. The maps and temperature/precipitation (T,P) graphs are redundant. Maps are not bringing any additional information since the modeling is too simplified to give realistic values (no map of grassland, spatial homogeneity of management practices, no fertilization & irrigation, no water feedback) except for some reader who are looking for specific values in some data. Maps can even be misleading. For example on fig.5b, the areas for which the difference is negative seem unrelated when in the T,P plots of fig. 6b, it is clear that there is a similar process in these regions due to their bioclimatic conditions. Moreover maps are difficult to compare visually to each other. They should be moved to appendix.

We take the point that with spatially homogeneous assumptions on management the resulting effects can be investigated best in relation to their climatic drivers as in the figures according to Fig. 2. Thus, we move the maps (formerly figs. 3, 5, 7, 10) to the appendix and concentrate on the T,P plots.

• To allow for mental representation of geographical distribution, the separation lines between T,P areas appearing in fig. 2 should be reported in later T,P plots to allow for rapid bioclimatic regions differentiation.

We appreciate this comment and will include black lines denoting the regions into the T,P plots.

• Also, even with maps in appendix, because maps and T,P plots show the same information (even if aggregated by deciles of precipitation and temperature in the latter) it would help the reader to use the exact same color scale for both families of plots showing a similar variable. The sequential green/blue color scale used
for T,P plots in fig 4 is less likely to introduce an artificial visual bias than the divergent color scale in fig 5.

We acknowledge this comment and use the color scale of Fig. 4 also for those maps which show actual values and not differences (formerly figs. 3a, b, c, 5a, 7a, 10a).

• In several occasions in the text, grass yield and soil carbon patterns are explained from their relation to NPP. The described relationship is not visible from the data shown making the text impossible to follow for the reader. Graphs of yield versus NPP and soil carbon versus NPP would help convince the reader of the significance of the trends and relationships described.

The trade-off between the harvesting or sequestration in the soil of assimilated carbon is not very simple to illustrate. We tried to find a way for the visualization (Fig. 1) and came up with a possibility to distinguish average harvest in relation to soil carbon and NPP in 2 ways. For a low livestock density (upper row), the ratio of harvest to the feed demand is only lower than 97% (or 0.97 in the figure) when NPP is below 50 gC m-2 a-1. At medium NPP values, very high soil carbon values occur but for most of the grid cells (dark color in upper right plot) soil carbon values are below 20 kgC m-2. At high grazing pressure (lower row), the demand cannot be fulfilled when NPP is below 400 gC m-2 a-1 and the occurrence of soil carbon values above 40 kgC m-2 is less often. In case, this diagram is helpful, we will include this representation in the results section.

Color scale used in the difference plots is counter-intuitive with increase in cold colors and decrease in warm colors. Also, the color scale is too close from the one with absolute values to see right away that what is plotted is a difference. We appreciate the attention paid to the presentation of the results but in this case, we keep the chosen view. When temperatures are displayed, the connection to 'cold' and 'warm' colors is fair but usually blueish colors are perceived as positive and reddish as negative. Thus, we have chosen this (color-blind safe) combi-
nation for the presentation of the differences. And we think that the same color scale for the actual values match very well with the positive values in the difference plots which can be distinguished easily by the color legend beside each figure.

- Fig. 5a and 10a are not described, what is described and should be shown (in appendix) is the difference in harvest between scenarios for consistency with text. Thus, harvest maps for options M,  $G_D$  and  $G_R$  (formerly figs. 5a, 7a, 10a) are exchanged by difference maps and included in the appendix.
- The authors attempt to compare their simulation to regional data in Europe. This exercise is very ambitious given the level of simplification of the model, in the spatial homogeneity, diversity of scenarios and processes involved. However it can be an informative comparison if well explained and described. For example, the reason for choosing to compare only the highest harvest GD simulations with data is not explained. If it is supposed to be the more realistic given current practices it should be justified. An interesting result would be to show which management setting leads to the best simulation in each subregion and try to explain it.

We take this comment as a reason to revise the description of the comparison and to include a new figure. Section 3.2 is now following a reasoning including 3 hypotheses and concentrates on 3 selections from the simulation results to test these. The section heading will be 'Comparison to harvest data' and the motivation will be extended to:

'Since management assumptions for the simulations were spatially homogeneous and management in Europe is known to vary spatially as well as temporally, we use the comparison to find out whether climate- and managementinduced variations in grass harvest can be captured by the applied options. Therefore, we formulate and test the hypotheses that European grass harvest GMDD
- 1. can be achieved by grazing animals only,
- 2. is determined by management and only to a minor degree by climate and
- 3. per geographical entity, a dominant management option can be identified that results in similar harvest values as reported.'

**The text for hypothesis 1 will use Fig. 14 b with**

'For testing hypothesis 1, we choose for each regional value from the simulation results for option  $G_D$  with varying livestock densities the maximum harvest value (see section ??, Fig. ?? b) which resulted mostly from simulations with medium stocking densities. Clearly, the pattern differs from the reported yield estimates (Fig. 14 a). The gradient of reported yields from northeast to southwest is underestimated and yields are higher in southern Europe and lower in the western parts of the continent. Thus, a continuous withdrawal of leaf biomass could achieve higher grass harvest in eastern Europe and the Mediterranean whereas for yields in western Europe (esp. Great Britain, The Netherlands and Norway) much higher values are reported than simulated. Therefore, we can reject hypothesis 1 and, thus, support the assumption that grassland management in Europe is not homogeneous concerning the presence of animals on the pasture or the harvesting intensity.' For hypothesis 2, Fig. 14 c is used:

'Hypothesis 2 cannot entirely be resolved with our simulation results but we test whether the reported gradient from northwest to southeast can be reproduced. From simulations with option  $G_D$ , we select per region the harvest flux closest to the reported values (Fig. 14 c) so that the livestock densities can be inferred that lead to the observed harvest values. The resulting pattern matches the reported values below 260 gC m-2 a-1 which are occurring in most of southern and eastern Europe as well as Scandinavia apart from Norway. Values are only underestimated in Great Britain, The Netherlands, Ireland and Norway on highly managed grasslands, e.g. which are fertilized and irrigated. Comparing those regions in which maximum values (Fig. 14 b) are the closest to reported values
(Fig. 14 c) but still more than 10 % too low, apart from the 4 countries mentioned before, German and French provinces appear. We interpret this as strong indicators for intensively managed systems. On the other hand, regions in which maximum harvest by grazing overestimates reported values by 50 % are located mainly in East European countries (Slovakia, Montenegro, Macedonia, Lithuania, Hungary, Croatia, Estonia, Belarus, Bulgaria, Albania). There, the potential of grass production is by far not utilized and climatic conditions are not limiting. Hypothesis 2 can be confirmed by these findings.'

The inclusion of Fig. 2 (as Fig. 14 d) allows to relate to hypothesis 3 with: 'For testing hypothesis 3, for each geographical region the closest value from all available simulation results is chosen (Fig. 14 d). The derived values deviate from those in Fig. 14 c in the highly managed countries identified before (Great Britain, The Netherlands and Ireland) and some regions in Finland, Germany and France. The options that result in closer values are the default option D and mowing M both describing pasture regimes with additional harvesting to increase yields. Regions in which reported yields are higher than in Fig. 14 d can thus be identified as definitely under regimes including other yield increasing measures such as fertilization and irrigation. Therefore, also hypothesis 3 can be justified by analysing the simulation results.'

 About the results in general, the article lacks an analysis of the results. We hope that with the inclusion of paragraphs on the underlying processes and their implementation in the model as well as with the changes in the comparison to harvest data (section 3.2) and in the derivation of potentials (section 3.3), the necessary analysis of the results is improved.

**Discussion**

• The discussion section is about effects on soil carbon, uncertainties and further developments in the modelling approach. If all these discussion points are in-
teresting, after a very descriptive results section, the reader is also expecting an interpretation of the results, as a full discussion with explanation of the underlying processes and implications. For example, what processes in the model drive the feedbacks? Some of the feedbacks are the simple expression of the relationships coded in the model and this should be identified and its realism described. What is the reason for the pattern in fig. 6,8 & 11 (climatic area 10

We delete the sentence within the new description of the results.

are the same. Rephrase, maybe use 'homogeneous'.

• *Fig. 9 please make the figure visually lighter by using only one color bar per row.* We take this point and will adjust the figures.

 P13L4 the sentence 'average grass yield and soil carbon under these conditions are not substantially different' is confusing. It sounds like grass yield and soil C

P17L21 rewrite sentence

The sentence is changed and split into two:

'Particularly high losses of soil carbon are simulated in cold regions (annual mean temperatures below 0 °C) (Fig. 9 c). There, soil carbon for low stocking densities (0.4 LSU ha-1) decreases by 2.5  $\pm$  2.8 kgC m-2 and for higher densities (2 LSU ha-1) even by 19.9  $\pm$  8.2 kgC m-2.'

• p19L11 not clear, rewrite sentence

This sentence was deleted when section 3.2 was completely rewritten.

p19L16 'low correlations', 'high standard deviations and RMSD'. Give the numbers.

Also this sentence will be deleted when section 3.2 is completely rewritten. Numbers for standard deviations and correlation coefficients from the Taylor diagram are then included in the description of Fig. 13.

P6L3 sentence not clear

We assume that the lack of clarity refers to the 'stand' and include a reference to the definition of the stand concept at the end of the sentence with '(concept see 2.2)'.

 P6l28 'are used' instead of 'is used' Plural form is corrected. GMDD
• P22L05 why compare LSUmax to scenario M and not default D as in all the rest of the manuscript?

The reasoning for this comparison is given in section 2.6.2 which is extended to: 'This comparison was chosen because under the mowing option neither biomass removal is maximized nor is harvested carbon added to the soil so that a rather moderate impact on soil carbon stocks is expected compared to grassland without harvest.'

**References**

- Chang, J., Ciais, P., Herrero, M., Havlik, P., Campioli, M., Zhang, X., Bai, Y., Viovy, N., Joiner, J., Wang, X., Peng, S., Yue, C., Piao, S., Wang, T., Hauglustaine, D. A., Soussana, J.-F., Peregon, A., Kosykh, N., and Mironycheva-Tokareva, N.: Combining livestock production information in a process-based vegetation model to reconstruct the history of grassland management, Biogeosciences, 13, 3757 3776, doi:10.5194/bg-13-3757-2016, 2016.
- Robinson, T. P., Wint, G. R. W., Conchedda, G., Van Boeckel, T. P., Ercoli, V., Palamara, E., Cinardi, G., D'Aietti, L., Hay, S. I., and Gilbert, M.: Mapping the global distribution of livestock, Plos One, 9, e96 084, doi:10.1371/journal.pone.0096084, 2014.
- Saitoh, T. M., Nagai, S., Noda, H. M., Muraoka, H., and Nasahara, K. N.: Examination of the extinction coefficient in the Beer-Lambert law for an accurate estimation of the forest canopy leaf area index, Forest Science and Technology, 8, 67 – 76, doi:10.1080/21580103. 2012.673744, 2012.
- Sitch, S., Smith, B., Prentice, I. C., Arneth, A., Bondeau, A., Cramer, W., Kaplan, J. O., Levis, S., Lucht, W., Sykes, M. T., Thonicke, K., and Venevsky, S.: Evaluation of ecosystem dynamics, plant geography and terrestrial carbon cycling in the LPJ dynamic global vegetation model, Global Change Biology, 9, 161 – 185, 2003.
- Smit, H. J., Metzger, M. J., and Ewert, F.: Spatial distribution of grassland productivity and land use in Europe, Agricultural Systems, 98, 208 219, doi:10.1016/j.agsy.2008.07.004, 2008.

---

## Author Response (AR2)

We thank the reviewer for the concise review and the remarks which we found to the point. We address the mentioned items one by one:

1) P4L5: from the detailed description in Methods, M is not an 'intensive' mowing.
   We see the point and exchange 'intensive' by 'regular' (P4L6).

2) P5L9: It might be better to cite original reference for the pasture area.
   Fader et al. (2010) modified the underlying dataset by Portmann et al. (2010) so that we changed the sentence (P5L9) to
   'as weights which was derived by Fader et al. (2010) by modifying the dataset of Portmann et al. (2010)'.

3) P6L26: LPJmL3.5 is not mentioned before. Either give a reference or a web link.
   We introduce the version before implementing the grassland options at P5L13 with
   'We refer to the current status of the model prior to the implementation of managed grasslands as LPJmL3.5.'

4) P8L14: small ruminants
   We changed 'smaller' to 'small' in P8L15.

5) P8L15: It still be necessary to define or mention where the combination management to be used. For example, rangeland may not be mown in most regions.
   We included this aspect by changing the sentence P8L17:
   'While most rangelands are not cut at all, grasslands can be grazed by high livestock densities for short time periods but rarely mowed.'

6) P8L24-26: the description here is not the same as in Sec. 1.2.
   With the implementation of the grassland options, we also changed the former default option which is now mentioned at P8L26 by writing
   'Under the improved harvest scheme'.

7) P9L3: Vegetated period could be much shorter than 1 year, which makes harvest may be failed in non-growing season? I am not convinced that option M is a good/typical management schemes. But it might be useful to represent in the future a combination of mowing+grazing.
   We adapted the description of this option to better motivate the current version. Indeed, there is room for development for this option which could be linked to local conditions. Now the description on P9L4 states:
   'The mowing option $M$ represents a regime with several mowing events per year. This option may be adjusted to local conditions by scheduling these events to certain dates or according to climatic conditions. For the global application, two harvest events per year are scheduled six months apart with one event on June 1$^{st}$ and another one on December 1$^{st}$. In so doing, the mowing option is identical on both hemispheres but can fail for events not within the growing period.'

8) P10L16: Is there a fixed fraction in the model for the carbon to animal and mineralized?
   We only regard the partition of harvested carbon between the soil and the rest and added the sentence on P10L20:
   'We do not distinguish the portions of the grazed carbon that is going to animals or to mineralization.'

9) P11L14: Why use 390-year spinup?

> The second spinup phase using the historical landuse development should ensure a near-equilibrium status for the transient run from 1901. The specific value of 390 years is derived from experience. We add the sentence on P11L17:
> 'Nearly twice the length of the landuse history of 200 years from 1700 to 1900 is needed to achieve consistent starting conditions for the transient simulations after the dynamic soil carbon equilibrium under potential natural vegetation has been disturbed.'

10) P11L27: why use 1998-2002 rather than usually used long-term mean? Please specify the reason.

> For this analysis we wanted to see climate related responses and averaging over longer periods would rather reflect the effects of other drivers than weather. We specify this with the insertion on P11L30 of
> 'representing a medium-range period that reflects weather-related phenomena without relying on single years.'

11) P12L9: harvest here is a little confusing. May think about change a word. biomass use potential?

> We agree that the word 'biomass use' fits the purpose better and exchanged it on P12L12, P1213 and P12L16 and in section 3.3.

12) P13L23: Here savanna is used. Does it fit the classification of Koppen classification? As mentioned by Reviewer 1, I cannot find the rationales behind this climate classification. It is not as mentioned in the response can help explain 'motivated by the values in the climate response figures', because the difference response to management did not well locate in the bioclimatic regions defined here.

> We see that the use of biome classifications are not adequate here. The purpose of the figures was to better describe the influence of the climatic conditions. We did not refer to the Köppen-Geiger classification which distinguishes between different seasonal developments because this cannot be illustrated in the way we intended to do. The classification within the figures was chosen after analysis of the results and has no connection to other classification schemes. Therefore, we removed any biome reference and restrict the description on the temperature range and humidity or aridity. The sentence on P14L1 is changed to:
> 'Arid and moderately warm regions ($T_A$ above 15°C and $P_A$ below 500 mm) are characterized by low NPP (< 200 gC $m^{-2} a^{-1}$) and grass yield (33 ± 36 gC $m^{-2} a^{-1}$) which corresponds to 46 % of the NPP.'

13) P13L27: It is a little hard to believe 72% of NPP is harvested (include belowground NPP?) given the curve in Fig. 1.

> The interaction of medium growth and high leaf biomass over a long vegetation period can indeed derive these high percentages. When the daily biomass increment by NPP is in the order of the harvested leaf fraction shown in figure 1 over the entire year, even higher percentages of the NPP could be harvested. We try to motivate this result by including the sentence on P14L4:
> 'This can be the case when the biomass increment by NPP is similar to the harvested biomass for a longer time period.'

14) P15L5: If the exponentially grow is due to lr, it would be necessary to explain other factors that should affect the regrowth such as nutrient limitation.

> For this feedback, the ratio lr does not play a role. Under the conditions mentioned here, the growth relative to the leaf carbon increases and - even more important – the maintenance respiration decreases so that the relative growth becomes higher. We include the sentence on P15L14:
> 'The moderate reduction in leaf carbon still ensures high productivity while reducing maintenance respiration so that the net increase in carbon grows overproportionally.'

15) P16L2: It seems not 'below' 500 in Fig 5.

> Indeed, the separation of the regions does not meet the description so that we exchanged the value 500 with 800 mm $a^{-1}$ on P16L1.

16) P16L11: 3.5 is not slightly higher in mean.

> Yes, we removed the word 'slightly' on P16L10.

17) P16L16: in previous similar presentation, 100 gC was used. Might be better to stick to it.

> We see the point and adjusted the values on P16L14-15 and also on P19L4.

18) P19L4-5: It may be necessary to explain this ratio under the context of turnover time of soil, since mowing/grazing only affect biomass input into the soil.

> Actually, the export of carbon via harvest as well as the additional flux into the soil play a role for the difference in the accumulation in the soil between the mowing and grazing options. We add 2 sentences on P19L7 including the values for the grazing option without livestock to illustrate the natural accumulation background without the additional flux into the soil:
> 'In comparison to the mowing options $D$ and $M$, the loss of carbon from the system by harvest is reduced so that the transfer of NPP into the soil leads to a higher accumulation in the soil per NPP. In the case of a livestock density of 0 LSU $ha^{-1}$, i.e. neither export of carbon via harvest nor additional transfer into the soil via manure, the accumulation results in 1560 gC per 100 gC NPP.'

19) P20L20-21: this sentence is hard to understand.

> We see the point and try to clarify by including 2 shorter sentences on P21L3:
> 'Grass harvest shows similar spatial patterns for both grazing options (compare Fig.A4 a and A3 a). Nevertheless, for option $G_R$ the required demand of grass harvest for the given livestock density is met on 42 % of the pasture area whereas the demand is met at 67 % of the area for option $G_D$.'

20) P25L4: 'animals are not only fed with grass' It is the situation for many regions of the world. Is it?

> We are aware that the comparison of the derived potentials with actual livestock densities is not considering the feed composition which differs considerably between livestock production systems and agro-ecological zones. Here, we just wanted to highlight that the few regions with lower simulated $LSU_{harv}$ values than reported are exactly those in which 'roadside grazing' or 'occasional feed' have high values in the feed baskets. We include a specification on P25L11:
> 'where feed baskets for ruminant livestock contain only minor shares of grass (Herrero et al., 2013).'

21) P25L11: And might be more importantly depending on the carbon-nutrient interactions.
We appreciate the reference to the importance of nutrients in relation to carbon and have to mention that neither nitrogen nor phosphorus limitations were considered so far. Since the publication of the implementation of the nitrogen cycle into LPJmL is currently under review (von Bloh et al., https://www.geosci-model-dev-discuss.net/gmd-2017-228/), we can shortly investigate also this aspect. Nevertheless, at the specific sentence (now P26L1), nutrients are not of major influence. As specified in the next sentence, the $LSU_{harv}$ values are low in regions with either very low temperatures or low precipitation so that we did not change the text.

22) P25L27: and may also no external fertilization.
Since there was no line 27 on page 25, we assume that the comment refers to P27L27. Indeed, the chosen references report on areas without additional fertilization.  We mention this in the sentence P28L2:
'when soil management did not include additional fertilization.'

23) P3L24: you mentioned in Sec. 4.2 that Chang et al., 2016 has applied a model at global scale.
This is correct and we have to mention this recent development in the introduction as well. The sentence at P3L23 is changed to
'While the implementation of grazing and mowing is demonstrated at the European scale, a recent application is combining satellite-derived productivity and model simulations at the global scale to derive historical changes in grassland management (Chang et al., 2016).'